# CrypticBio: A Large Multimodal Dataset for Visually Confusing Species

**Georgiana Manolache[1], Gerard Schouten[1], Joaquin Vanschoren[2]**

[1]Fontys University of Applied Sciences, Eindhoven, The Netherlands
[2]Technical University of Eindhoven, Eindhoven, The Netherlands

Correspondence: `g.manolache@fontys.nl`

## Abstract

We present CRYPTICBIO, the largest publicly available multimodal dataset of visually confusing species, specifically curated to support the development of AI models in the context of biodiversity applications. Visually confusing or cryptic species are groups of two or more taxa that are nearly indistinguishable based on visual characteristics alone. While much existing work addresses taxonomic identification in a broad sense, datasets that directly address the morphological confusion of cryptic species are small, manually curated, and target only a single taxon. Thus, the challenge of identifying such subtle differences in a wide range of taxa remains unaddressed. Curated from real-world trends in species misidentification among community annotators of iNaturalist, CRYPTICBIO contains 52K unique cryptic groups spanning 67K species represented in 166 million images. Records in the dataset include research-grade image annotations—scientific, multicultural, and multilingual species terminology, hierarchical taxonomy, spatiotemporal context, and associated cryptic groups. To facilitate easy subset curation from CRYPTICBIO, we provide an open-source pipeline, CRYPTICBIO-CURATE. The multimodal design of the dataset provides complementary cues such as spatiotemporal context that support the identification of cryptic species. To highlight the importance of the dataset, we benchmark a suite of state-of-the-art foundation models across CRYPTICBIO subsets of common, unseen, endangered, and invasive species, and demonstrate the substantial impact of spatiotemporal context on vision-language zero-shot learning for cryptic species. By introducing CRYPTICBIO, we aim to catalyze progress toward real-world-ready fine-grained species classification models for biodiversity monitoring capable of handling the nuanced challenges of species ambiguity. The data and the code are publicly available in the project website[1].

## 1   Introduction

Advancements in AI are set to play a pivotal role in biodiversity conservation and ecological management as data in open citizen science platforms amasses. iNaturalist [28] and Observation.org [38] are well established citizen science platforms collecting biodiversity data worldwide, featuring annotated in-situ images of a wide range of species as well as metadata such as geographical location and observation date. AI-ready datasets are a crucial part of the development, evaluation, and eventual deployment of machine learning (ML) systems, and many studies have already demonstrated their potential for species identification [44, 47, 51]. As it turns out, however, species identification combines a unique set of challenges. As illustrated in Figure 1, these include: (1) viewpoint variations; (2)

---

[1]georgianagmanolache.github.io/crypticbio

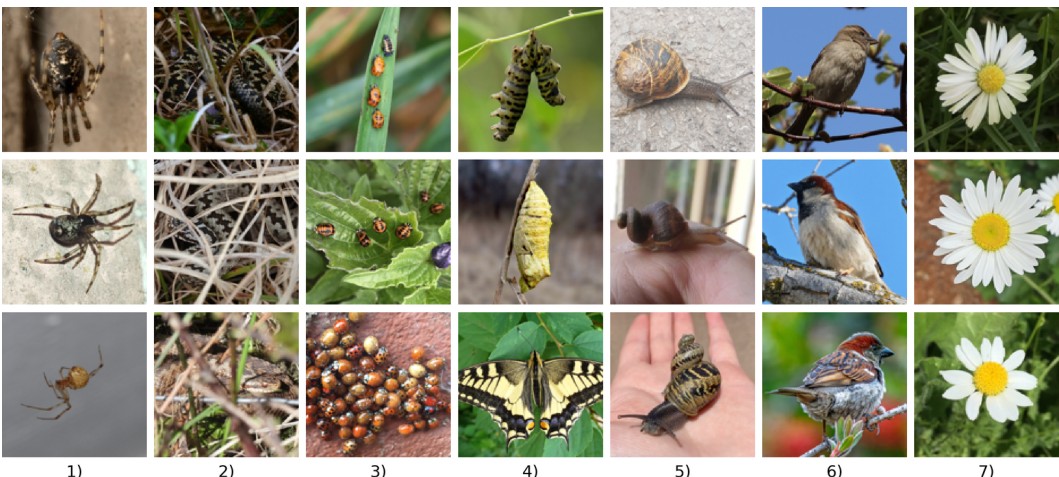

Figure 1: Challenges of biodiversity: (1) viewpoint variations (*Parasteatoda tepidariorum*); (2) occlusion by other objects (*Vipera berus*); (3) clutter (*Harmonia axyridis*); (4) multiple life cycle stages (*Papilio machaon*); (5) deformations (*Cornu aspersum*); (6) intra-class variation (*Passer domesticus*); (7) inter-class similarity (*Bellis perennis*, *Leucanthemum vulgare*, *Chamomile matricaria*).

occlusion by other objects; (3) clutter; (4) multiple life cycle stages; (5) deformations; (6) intra-class variation; (7) inter-class similarity. The latter two challenges are particularly hard: some species may have significant visual differences, while at the same time visual similarities in shape and color may exist between some species belonging to different classes. This morphological confusion makes it difficult even for humans to distinguish species without deeper level of expertise, and subsequently limits the construction of trustworthy AI for biodiversity [25].

Existing state-of-the-art datasets of text-annotated biodiversity images primarily curated from iNaturalist focus on taxa identification holistically. Notable examples include TREEOFLIFE-10M [47] with over 10 million observations spanning 451K species, and BIOTROVE [51] with over 161 million observations spanning 366K species, respectively. More recently, the multimodal dataset TAXABIND-8K [44] extending over 8K text-annotated biodiversity images with other contextual metadata like geographical location, environmental features, audio recordings, and satellite imagery, shows significant improvements on 2K bird species identification, using vision as the binding modality in a unified embedding space. Datasets that directly address the morphological confusion of groups of two or more species are significantly smaller, manually curated, and focused on a single taxon [3, 4, 34, 42]. Thus, the challenge of identifying subtle differences in a wide range of taxa remains to be addressed.

In this paper, we challenge the biodiversity AI research by curating and releasing **CRYPTICBIO**, a **multimodal dataset** comprising over **166 million images** of **52K unique visually confusing species groups** spanning **67K species**. Table 1 summarizes how CRYPTICBIO compares to prior biodiversity datasets in scale and annotation richness, and Table 2 contrasts it with existing cryptic species benchmarks (which are several orders of magnitude smaller). As this work is intended to have a direct impact on the use of AI for biodiversity research, we hope it will provide valuable insights to researchers seeking to better understand biodiversity. Our main contributions include: (1) the largest multimodal cryptic species dataset to date; (2) broad taxonomic coverage beyond prior single-taxon studies, (3) enriched metadata (locations, dates, multicultural and multilingual names) enabling new multimodal research; (4) an open-source pipeline for easy dataset curation (CRYPTICBIO-CURATE); (5) evaluation benchmarks demonstrating the utility of context in species identification.

The remainder of the paper introduces the CRYPTICBIO dataset and its relation to existing work (section 2), outlines the curation methodology (section 3), presents benchmark tasks and zero-shot results using CLIP-style models (section 4), and concludes with a summary and discussion of limitations (section 5). Supplementary material includes implementation details, extended results, and dataset access instructions.

Table 1: CRYPTICBIO comparable datasets.

| Dataset | Images | Species | Annotations | Source | Features |
|---|---|---|---|---|---|
| **CRYPTICBIO** | 166.5M | 67.1K | (multicultural and multilingual) vernacular scientific terms, taxonomic hierarchy, location, date, cryptic species group | GBIF (iNaturalist and Observation.org), GBIF Backbone Taxonomy [45], iNaturalist Taxonomy [30] | multimodal, cryptic species group (52.7K groups) |
| **BIOTROVE** [51] | 161.9M | 366.6K | vernacular, scientific terms, taxonomic hierarchy | iNaturalist | biased vernacular species terminology |
| **TREEOFLIFE-10M** [47] | 10.4M | 454.1K | vernacular, scientific terms, taxonomic hierarchy | iNaturalist, Encyclopedia of Life (EOL)[9], BIOSCAN-1M [22] | biased vernacular species terminology |
| **TAXABIND-8K** [44] | 8.8K | 2.2K | vernacular, scientific terms, taxonomic hierarchy, location, environmental features, audio recording, satellite imagery | iNaturalist, iNat2021[48], Santinel-2[6], WorldClim-2.1[10] | multimodal, bird species exclusive |

Table 2: Existing benchmarks; each represent one cryptic group. Our new benchmarks are described in section 4.

| Taxon | Benchmark | Images | Species | Annotations | Source |
|---|---|---|---|---|---|
| *Aves* | **AMAZON PARROTS** [34] | 14K | 35 | scientific terms | iNaturalist, eBird [8], Google Images |
| *Insecta* | **BUMBLE BEES** [46] (not publicly available) | 89K | 36 | scientific terms | iNaturalist, Bumble Bee Watch [24], BugGuide [2] |
| | **CONFOUNDING SPECIES** [4] (not publicly available) | 100 | 10 | scientific term | iNaturalist |
| *Mammalia* | **CHIROPTERA RHINOLOPHIDAE RHINOLOPHUS** [3] | 293 | 7 | scientific terms | personal collection during field surveys |
| *Reptilia* | **SEA TURTLES** [1] (not publicly available) | 6.9K | 36 | vernacular, scientific terms | Internet |
| | **SQUAMATA LACERTIDAE PODARCIS** [42] | 4.0K | 9 | scientific terms | personal collection during field surveys |

## 2  CRYPTICBIO Dataset

CRYPTICBIO comprises over **166 million images** from **52K unique cryptic groups** spanning **67K species**. Derived from **iNaturalist's records of historical misidentifications**, these groups are not symmetric, and their sizes vary, which explains the difference between the number of species and the number of cryptic groups (see details in the section 3). Figure 2 showcases cryptic species group examples over the representative taxa in biodiversity, while Figure 3 details the cryptic species group size distribution (see details in supplementary material B and D). The dataset is curated from **research-grade citizen science observations** provided by the Global Biodiversity Information Facility (GBIF) [11], containing validated data from iNaturalist, a source of demonstrated 95% annotation reliability [26], and Observation.org, an expert exclusive data validation source [12, 13, 15, 17, 16, 18–21]. In iNaturalist, observations are of research-grade quality at genus, species, or subspecies level under the following criteria: (1) at least two identifications (including the observer's); and (2) there is an agreement among at least two-thirds of identifiers [29]. Observation.org data is validated through: (1) a structured review process in which only expert validators assess the presence and quality of supporting multimedia; or (2) a computer vision system, available through the Nature Identification API (NIA) [40], validates with high confidence [41].

Images in CRYPTICBIO are annotated with detailed taxonomic descriptions and observation context, enabling extensive filtering and analysis. As summarized in Table 3, every observation at species level includes its scientific name and associated **six-tier taxonomic hierarchy** deterministically derived from GBIF Backbone Taxonomy [45].

While the inclusion of common or vernacular terminology alongside Latin binomials has been recognized as an important step toward accessibility and inclusivity in biodiversity datasets [47],

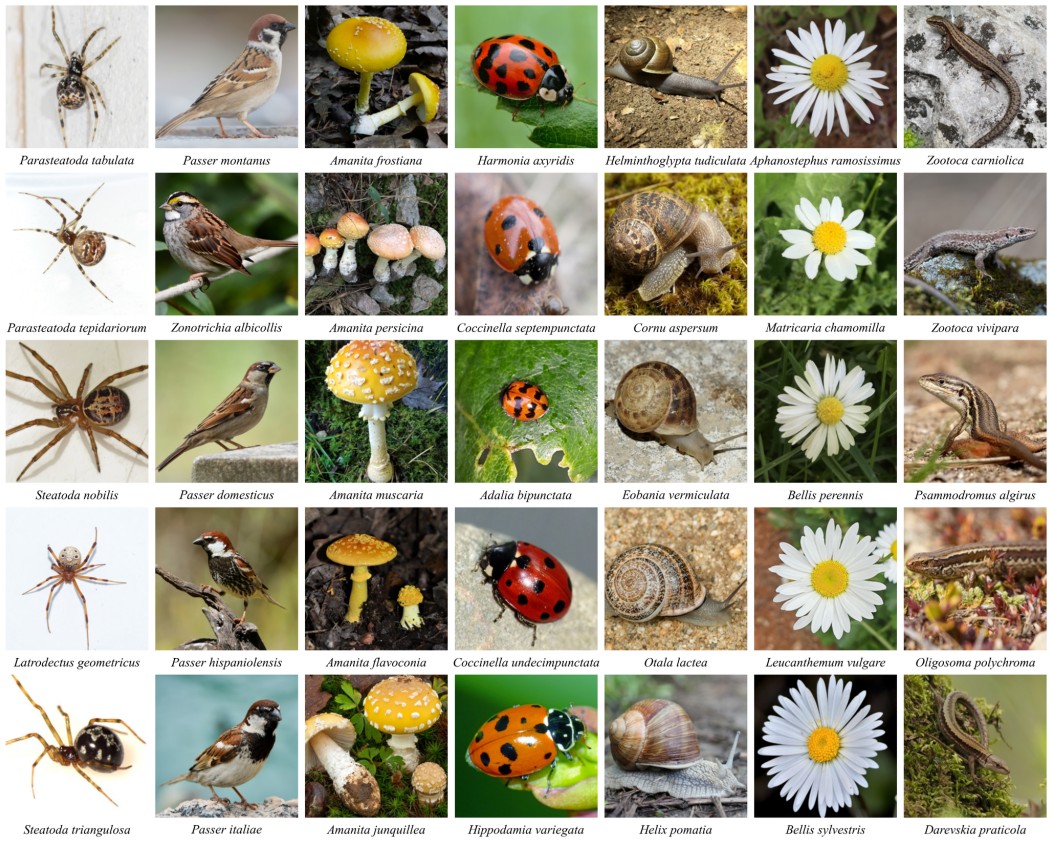

Figure 2: Example of cryptic species in CRYPTICBIO. Each column shows from left to right a cryptic group from *Arachnida*, *Aves*, *Fungi*, *Insecta*, *Mollusca*, *Plantae*, and *Reptilia*, taxa representative in biodiversity conservation and policy change supervision.

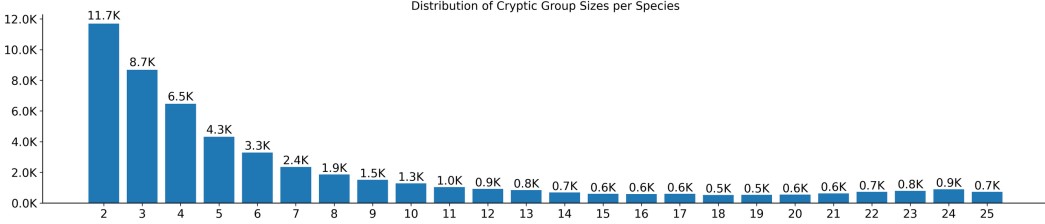

Figure 3: Cryptic group size distribution in CRYPTICBIO. The long-tailed distribution suggests that the majority are divided into a small number of cryptic entities.

relying solely on English risks marginalizing indigenous and culturally specific naming traditions. Moreover, even within the English-speaking world, regional naming conventions can introduce their own biases. For example, the species *Perisoreus canadensis* is commonly known as the *Canada Jay* in Canada, yet referred to as the *Gray Jay* in the United States [36]. Despite this variability, existing biodiversity datasets standardize to a single vernacular name per species, overlooking the cultural and linguistic diversity embedded in naming practices. We believe integrating **multicultural and multilingual species vernacular names** preserves ecological knowledge and equity, and increases inclusivity and cultural reach. CRYPTICBIO enumerates scientific and vernacular species terminology as listed in the iNaturalist Taxonomy Archive [30].

Building on prior work demonstrating that spatiotemporal priors improve species identification [5, 7, 43], we integrate **spatiotemporal context as an additional modality** which can then eventually be

Table 3: CRYPTICBIO annotations enumerate 15 fields provided in 627 Parquet formats; the dataset is openly available (for download and browsing) on HuggingFace Datasets.

| Type | Description |
|---|---|
| Species scientific name | Scientific species name (Latin binomial), represented as a string in field `scientificName`. |
| Species vernacular name(s) | Multicultural species common or vernacular names (i.e., *Perisoreus canadensis* is commonly referred to as the *Canada Jay* in Canada, while in the United States is referred to as *Gray Jay*), represented in a list of comma separated strings in field `vernacularName`. |
| Taxonomic hierarchy | Species (primary) taxonomic hierarchy deterministically derived from species scientific name, represented as strings in separate fields: `kingdom`, `phylum`, `class`, `order`, `family`, `genus`. |
| Date | The date when the species was observed (separated DD, MM, YYYY), represented in fields `day`, `month`, `year`. |
| Geographical location | Latitude, longitude coordinates where the species was observed (decimals), represented in two fields `decimalLatitude` and `decimalLongitude`. |
| Cryptic species group | One or more species misidentified with the focal species, noted by scientific name, represented in a sequence of strings in field `crypticGroup`. |
| URL | Downloadable image link from Naturalist and Observation.org repositories, represented as a string in field `url`. |

aligned in a common embedding space. Figure 4 details the spatiotemporal distribution of the dataset. Cryptic species have historically emerged as a consequence of biogeographic isolation (natural barriers, such as rivers, mountain ranges, or deserts; deforestation; agricultural expansion; or man-made structures) which disrupted gene flow between populations and ultimately promoted allopatric divergence over evolutionary timescales [25]. We hypothesize that the integration of temporal (date) and spatial (geographical coordinates) context provide complementary cues beyond visual appearance alone and ultimately enhance the identification of cryptic species. Figure 5 illustrates an example of two cryptic bird species that have distinct geospatial distribution patterns and it is easy to tell them apart based on the location.

Additionally, to support reproducibility and extensibility, we release the full data curation pipeline CRYPTICBIO-CURATE, enabling streamlined access, manipulation, and image download via raw URLs for CRYPTICBIO subsets curation.

**License** Only images released under a Creative Commons license (CC BY-NC 4.0) are included, ensuring that the dataset is openly available for public research and non-commercial use.

**Geoprivacy** We include geolocation metadata for all records, relying on the source platforms' automated and user-specified obscuration for sensitive species [27, 39]. This means endangered or protected taxa have deliberately imprecise coordinates, in line with geoprivacy best practices.

**Offensive content** We opted not to remove occasional graphic images (e.g. predation or roadkill) to maintain ecological authenticity. These instances are infrequent, but we advise users—especially when deploying models or visualizations—to be mindful that some images may be upsetting to general audiences.

**Responsible use** Models trained on this data should not be used for unlawful wildlife tracking or poaching; we provide the data to support conservation efforts and ecological research, aligning with NeurIPS ethical guidelines.

**Privacy** We strictly exclude all personally identifiable information (PII) from the metadata associated with the dataset, ensuring that fields such as observer names and email addresses are removed. However, we acknowledge that in rare cases, PII may still be visible within the image content itself; for example, faces of individuals, vehicle license plates, distinctive property features, or GPS location markers embedded in the media. While such occurrences are unintended and infrequent, users of the dataset should be aware of this residual risk when analyzing or displaying images.

In comparison to prior biodiversity datasets (e.g., TREEOFLIFE-10M [47] and BIOTROVE [51], see Table 1) that take a broad but coarse approach, CRYPTICBIO is the first to specifically target cryptic species at massive scale, with 166M images across 52K cryptic groups—significantly enriching each observation with spatiotemporal context and multicultural vernacular names. Compared to the only

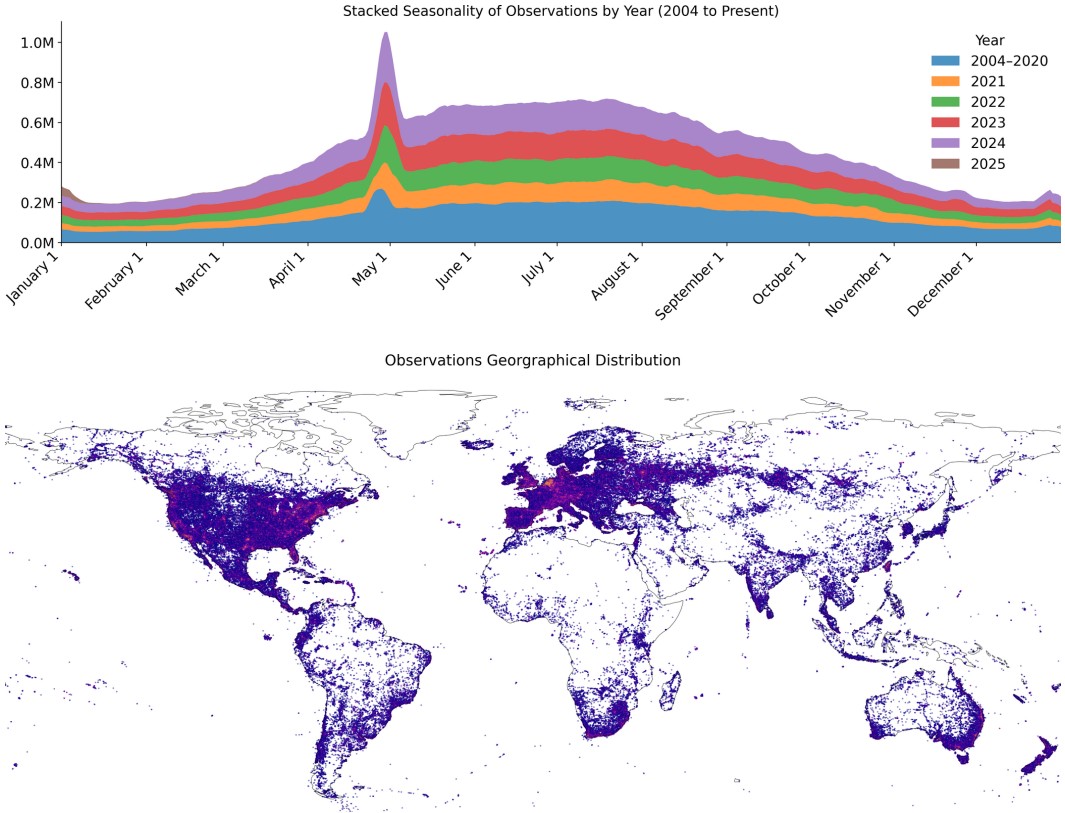

Figure 4: Spatiotemporal distribution of CRYPTICBIO: (top) stacked seasonality distribution; (bottom) geographical distribution. Majority of records are concentrated in Europe and North America, with a seasonal peak in observations during May.

other multimodal dataset TAXABIND-8K (focused exclusively on 2K bird species), CRYPTICBIO covers 67K species across diverse taxa with vision+language+spatiotemporal modalities, making it a general, real-world AI-for-biodiversity benchmark.

## 3  Data Curation

**Cryptic species challenge** Cryptic species groups are derived from iNaturalist "Similar Species" tab on a species profile page. When an observation originally identified as species A is later reclassified as species B, species B is designated as a commonly misidentified counterpart of A. These empirically derived confusion links, aggregated across many observations, determine which taxa appear in each other's "Similar Species" lists. Importantly, these groups are not symmetric; if species A is often confused with B, C, and D, species B's "Similar Species" does not necessarily include C or D as common misidentifications. This data-driven cryptic species group curation reflects real-world trends in species misidentification among community annotators. By leveraging this organically generated confusion structure, we design a dataset that reflects the practical challenges of fine-grained species identification—particularly in cases where even skilled annotators struggle to distinguish between species.

Because this information is not exposed through the iNaturalist API or bulk exports, CRYPTICBIO relies on a reproducible scraping workflow to extract the "Similar Species" tab content directly from public species pages. Importantly, this procedure respects iNaturalist's `robots.txt` policy, targets only non-restricted endpoints, and introduces a 20-second delay between requests to avoid excessive server load. To ensure transparency and reproducibility, we release the full scraping code, metadata, and curated outputs in GitHub, allowing others to replicate the construction of cryptic species groups under the same conditions. While we recognize the limitations of this approach, it provides the only

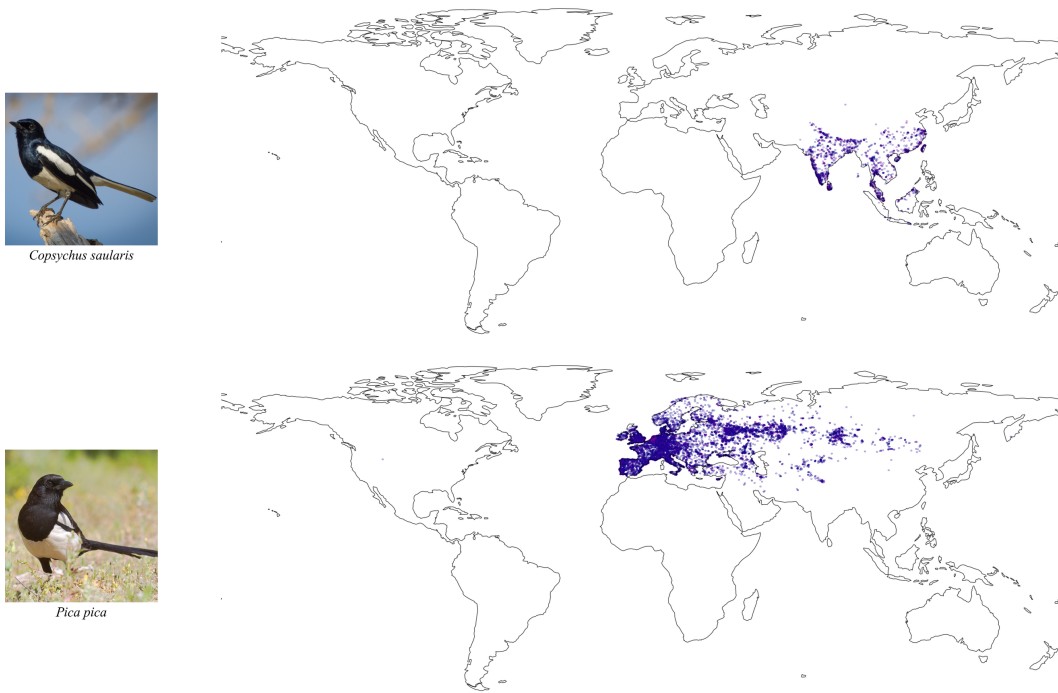

Figure 5: The importance of geospatial information demonstrated by two visually similar species and their distinct patterns in geospatial locations from CRYPTICBIO.

current means of systematically capturing iNaturalist's confusion-based links, and we are actively investigating API-based or officially supported alternatives for future iterations of the dataset.

**Curation and filtering** To assemble a taxonomically diverse dataset utilizing GBIF's data portal, we apply a series of structured filters to occurrence records. GBIF comprises over 217M occurrences as of 2025-04-13, originating from citizen science sources iNaturalist Research-Grade Observations and Observation.org, stored in a Darwin Core Archive standard vocabulary (CSV files `occurrence`, `multimedia` and additional metadata `eml.xml` and `meta.xml`). We select observations of most frequent taxa from *Animalia* (classes *Arachnida*, *Aves*, *Insecta*, *Mollusca*, *Reptilia*), *Plantae*, and *Fungi* kingdoms, and join each `occurrence` and `multimedia` files, discarding irrelevant columns. We filter observations CC licensed image files (bird observations may include audio or video media files) made only species level and enrich them with primary taxonomic hierarchy levels (kingdom, phylum, class, order, family, and genus [31]), as well as multicultural English vernacular species terminology from iNaturalist Taxonomy [30]. We retain the temporal (date) and spatial (latitude and longitude) metadata associated with each occurrence as additional contextual information. The associated images are referenced by downloadable URLs, ensuring direct access to the visual media for each observation (see details in supplementary material C).

Cryptic groups are structured using species scientific terminology based on scraped information from iNaturalist, and stored in JSON format. After extracting these groups, we merge them with GBIF observations, ensuring that we also retain species that do not have their own cryptic group but are listed as members of another species' group. Finally, the data are exported in the Parquet format and made publicly available on HuggingFace Datasets.

As outlined, we release CRYPTICBIO-CURATE, a configurable preprocessing pipeline that streamlines the preparation of biodiversity datasets for multimodal learning. The pipeline (1) loads raw metadata (e.g., species scientific names, image URLs); (2) applies customizable filters to ensure data quality (such as balancing class distributions); (3) downloads associated images; and (4) outputs a curated dataset in a standardized format.

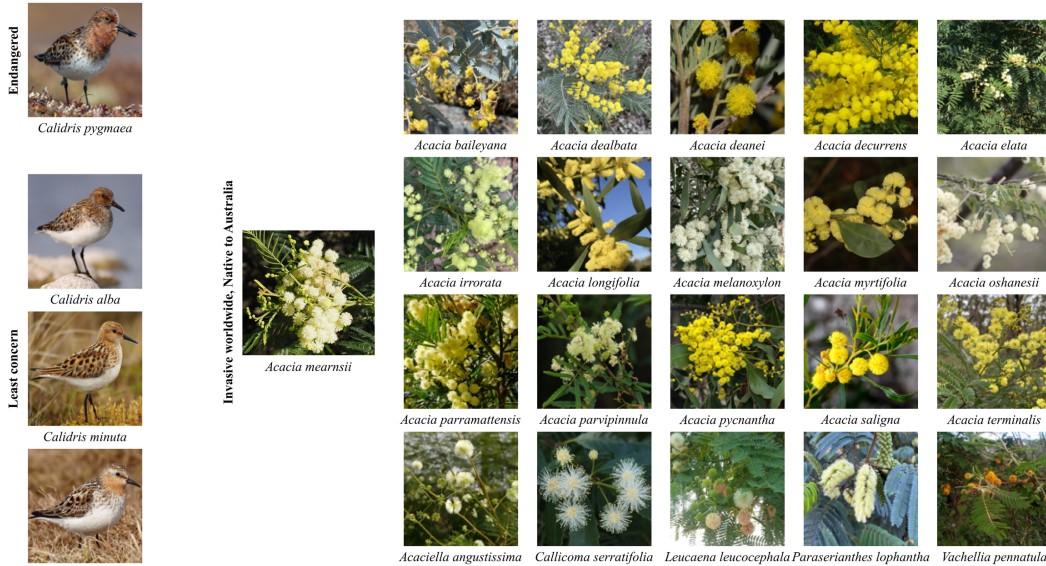

Figure 6: Example images from CRYPTICBIO benchmarks: (left) CRYPTICBIO-ENDANGERED *Calidris pygmaea* cryptic species group; (right) CRYPTICBIO-INVASIVE *Acacia mearnsii* cryptic species group.

## 4 Benchmarks

### 4.1 New Benchmarks

We curate four new cryptic species benchmark datasets, complementary to existing benchmark listed in Table 2. Figure 6 illustrates example images from our new benchmarks, which include both endangered and invasive cryptic species. We rigorously balance species distribution to enable more reliable and equitable cryptic species group identification (see new benchmark details in supplementary material E): for all our benchmarks we randomly select 100 samples from each species in a cryptic group where there are more than 150 observation per species.

**CRYPTICBIO-COMMON** We curate one common species (species with >10K observations) from *Arachnida*, *Aves*, *Insecta*, *Plantae*, *Fungi*, *Mollusca*, and *Reptilia* and associated cryptic group, spanning n=158 species.

**CRYPTICBIO-COMMONUNSEEN** To assess performance on common species from CRYPTICBIO-COMMON not used during model training, we specifically curate a subset containing data from 01-09-2024 to 01-04-2025, spanning n=133 species. By doing so, we ensure there are no duplicated images in training and inference.

**CRYPTICBIO-ENDANGERED** We propose a cryptic species subset of global IUCN Red List [32] endangered species. We select one endangered species from *Arachnida*, *Aves*, *Insecta*, *Plantae*, *Fungi*, *Mollusca*, and *Reptilia* and associated cryptic group, spanning n=37 species.

**CRYPTICBIO-INVASIVE** We also propose a cryptic species subset of invasive alien species (IAS) according to global the Global Invasive Species Database (GISD) [23]. IAS are a significant concern for biodiversity as their records appear to be exponentially rising across the Earth, and their geographical context is crucial [37]. We select one invasive species from *Aves*, *Fungi*, *Insecta*, and *Plantae* and associated cryptic group, spanning n=72 species.

### 4.2 Experiments

**Models** We evaluate state-of-the-art CLIP-style models trained on biodiversity data using the scientific and vernacular terminology of species. We use BIOCLIP [47]; BIOTROVE's BIOCLIP ViT-B-16 and OpenAI ViT-B-16 fine-tuned variants [51]; and TAXABIND [44] as image-only baseline models.

Table 4: Zero-shot learning on various models and benchmarks: I / L / E refers to image / location / environmental features embeddings; AP refers to AMAZON PARROTS [34] n=35 species; SLP refers to SQUAMATA LACERTIDAE PODARCIS [42] n=9 species; CRR refers to CHIROPTERA RHINOLOPHIDAE RHINOLOPHUS [3] n=7 species; CB-C refers to CRYPTICBIO-COMMON n=158 species; CB-CU refers to CRYPTICBIO-COMMONUNSEEN n=133; CB-E refers to CRYPTICBIO-ENDANGERED n=37 species; CB-I refers to CRYPTICBIO-INVASIVE n=72 species; WA refers to weighted average; BC refers to BIOCLIP; BT-B refers to BIOTROVE-CLIP-BIOCLIP; BT-O refers to BIOTROVE-CLIP-OPENAI; TB refers to TAXABIND. We mix scientific and common terminology were avaiable.

| Model | | AP | SLP | CRR | CB-C | CB-CU | CB-E | CB-I | WA |
|---|---|---|---|---|---|---|---|---|---|
| **BC** | I | 18.1 ±1.26 | 14.3 ±1.14 | **36.1 ±1.57** | 42.7 ±1.61 | 45.7 ±1.63 | 51.1 ±1.63 | 49.1 ±1.63 | 44.36 |
| **BT-B** | I | 14.1 ±1.14 | 14.1 ±1.14 | 16.0 ±1.20 | 58.9 ±1.61 | 61.6 ±1.59 | 45.8 ±1.63 | 58.3 ±1.62 | 51.54 |
| **BT-O** | I | **30.1 ±1.50** | **25.8 ±1.43** | 19.4 ±1.29 | 48.6 ±1.63 | 49.8 ±1.55 | 41.1 ±1.61 | 48.9 ±1.63 | 44.60 |
| **TB** | I | 15.1 ±1.17 | 14.8 ±1.16 | **36.1 ±1.57** | 46.1 ±1.63 | 48.8 ±1.63 | **52.4 ±1.63** | 52.2 ±1.63 | 46.73 |
| **TB** | I+L | - | - | - | 46.2 ±1.63 | 49.0 ±1.63 | **52.4 ±1.63** | 52.3 ±1.63 | 49.77 |
| **BT-B** | I+L | - | - | - | **61.9 ±1.58** | **64.2 ±1.56** | 45.2 ±1.63 | **63.2 ±1.57** | **58.14** |
| **BT-O** | I+L | - | - | - | 48.8 ±1.63 | 51.7 ±1.63 | 40.8 ±1.60 | 50.5 ±1.63 | 47.98 |
| **BT-B** | I+E | - | - | - | 25.9 ±1.43 | 26.1 ±1.43 | 33.1 ±1.61 | 30.1 ±1.50 | 28.65 |
| **BT-O** | I+E | - | - | - | 20.9 ±1.33 | 22.3 ±1.36 | 30.1 ±1.50 | 24.5 ±1.41 | 24.48 |

For multimodal learning, we add embeddings obtained from the image encoders to those obtained from TAXABIND location and environmental features encoders, which are then used for zero-shot classification. We collect from WorldClim-2.1[10] environmental features for each observation based on the location metadata, which are then passed through TAXABIND's environmental encoder.

**Metrics** We evaluate top-1 zero-shot accuracy across all new and existing benchmarks. We include a 95% confidence intervals for all reported metrics, calculated using binomial proportion confidence interval method (denoted as $\pm$). Furthermore, we compute an aggregate performance metric, which represents the weighted average accuracy over all classes across the benchmarks. Datasets used in our benchmark is balanced by design, with an equal number of samples per class (n=100). As a result, macro- and weighted-average metrics are effectively equivalent and are not affected by class imbalance. To assess the significance of pairwise performance differences between models, we use McNemar's test (p-value < 0.05).

**Results** Table 4 reports the performance on various benchmarks (for more details, see supplementary material F). We find that location embeddings significantly improve model performance on zero-shot image classification for cryptic species (p-value < 0.05). While part of the observed gain from adding image and location embeddings likely reflects the real-world geographic separation of morphologically similar species, it is also possible that data collection is geographically biased, with certain species more frequently observed and labeled in particular regions. In such cases, location embeddings may act as proxies for latent biases in the training distribution, effectively anchoring predictions in more probable species given past observer behavior and sampling hotspots. However, in our setting, the evaluation datasets are randomly sampled from the full data distribution, without explicit regional or taxonomic filtering. This suggests that the performance boost is not merely an artifact of overfitting to spatial bias, but rather a reflection of how location embeddings can capture species ranges and statistical tendencies (i.e. regional observation frequencies) present in the broader data. We encourage the AI community to create new subsets of CRYPTICBIO for various regions and measure performance against current benchmarks.

We observe that incorporating environmental variables alongside images results in a significant drop in zero-shot accuracy. We believe this is due to the limited discriminative value of environmental features for fine-grained classification, particularly within cryptic species groups that often share similar habitats. Additionally, environmental embeddings may be coarse, noisy, or misaligned with the image modality, which can dilute visual signals in a shared embedding space. This effect is especially pronounced in zero-shot settings, where model robustness is sensitive to modality noise and fusion quality.

We find that larger cryptic groups are associated with better model performance (Spearman p-value > 0.05 and Pearson p-value > 0.07). Its worth noting species in larger groups tend to be more common (i.e., CRYPTICBIO-COMMON and CRYPTICBIO-COMMONUNSEEN) and overrepresented in public biodiversity datasets like iNaturalist and Observation.org, resulting in stronger image-text

associations even in zero-shot. Smaller cryptic groups often consist of rare or underrepresented species, which pose a greater challenge due to fewer images. Future benchmarks should consider stratifying evaluation by group size and representation level.

**Limitations** We recognize that this data-driven identification of cryptic groups may miss rarely observed lookalike species. In future expansions, incorporating expert knowledge or targeted sampling of under-reported taxa could help capture cryptic relationships that have not yet appeared in crowd-sourced data.

Although our dataset includes temporal metadata, we have not systematically evaluated model performance when explicitly embedding this modality. This limits our understanding of how well the model leverages temporal patterns. While we still report significant zero-shot results using pretrained embeddings derived from different biodiversity datasets, few-shot learning could further enhance performance by enabling task-specific adaptation with minimal supervision.

## 5   Conclusion

We introduce CRYPTICBIO, the largest publicly available multimodal dataset designed to better understand cryptic biodiversity and ultimately, accelerate trustworthy AI solutions for biodiversity. Curated from research-grade sources, this dataset focuses on visually confusing species groups with rich associated metadata, surpassing existing cryptic species datasets in scale by several orders of magnitude. Our benchmarking across CRYPTICBIO underscore the value of context-aware multimodal datasets for advancing foundation models in biodiversity research, particularly for challenging cryptic species. CRYPTICBIO is publicly available (for download and browsing) on HuggingFace Datasets, and we release a comprehensive pipeline (CRYPTICBIO-CURATE on GitHub) to facilitate custom subset creation and reproducibility. With CRYPTICBIO, we aim to accelerate the development of AI models that are equipped to handle the real-world nuanced and context-dependent challenges of species ambiguity.

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

# A Supplementary Material

# B Ethics statement

## B.1 Taxon selection

We select seven most representative taxa in biodiversity conservation and policy change supervision: *Arachnida*, *Aves*, *Fungi*, *Insecta*, *Mollusca*, *Plantae*, *Reptilia*, as shown in Figure 7. These taxa represent the majority (>70%) of threatened species (left) and harmful invaders (right), underscoring their significant ecological and economic impact. Figure 8 shows examples of top five most frequent species and their counts.

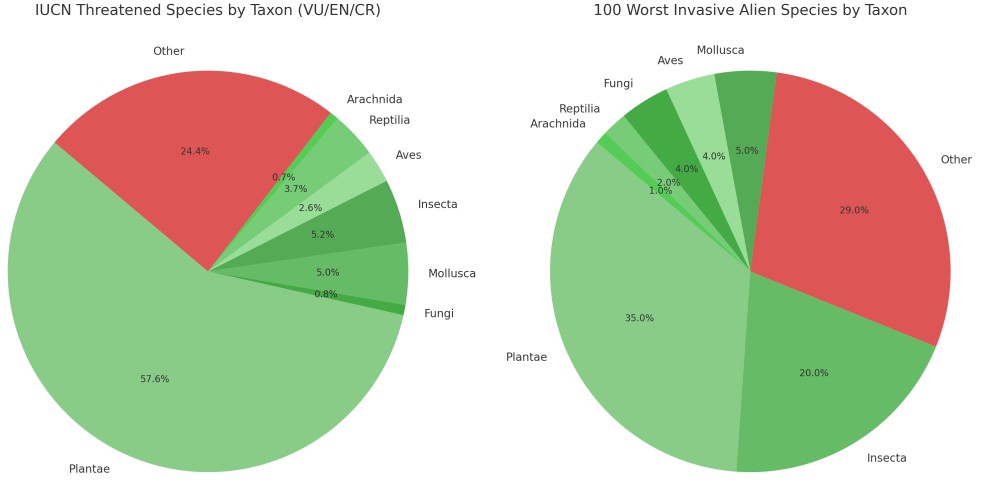

Figure 7: Taxa representativeness in biodiversity conservation and policy change supervision: (left) IUCN [32] endangered species distribution (labeled VU=vulnerable, EN=endangered, CR=critically endangered, the highest threats); (right) GISD [35] 100 worst alien species distribution.

## B.2 Endangered species location disclosure

It is critically important not to disclose the precise locations of threatened species because doing so can inadvertently put them at even greater risk. Many vulnerable species face threats from poaching, illegal wildlife trade, habitat disturbance, and over-collection. Sharing exact geographic coordinates, especially online or in open databases, can make it possible to locate and exploit these species.

To mitigate the risks associated with the disclosure of sensitive biodiversity data, citizen science platforms iNaturalist and Observation.org implement automatic geoprivacy measures for taxa listed on the global IUCN (International Union for Conservation of Nature) Red List of Threatened Species [32]. Our dataset contains less than 2.3K endangered species according to IUCN, as shown in Figure 9.

## B.3 Invasive alien species require geographical context

Accurate identification of visually similar species is critical, particularly when distinguishing between invasive and non-invasive taxa. Many invasive species closely resemble native or benign taxa. Additionally, geographic information plays a critical role in this process, as the impact of a species can vary by region—what is considered invasive in one area may be benign or even native in another. Overall, distinguishing invasive species within the appropriate geographic context is a foundational step in safeguarding biodiversity and maintaining ecological resilience.

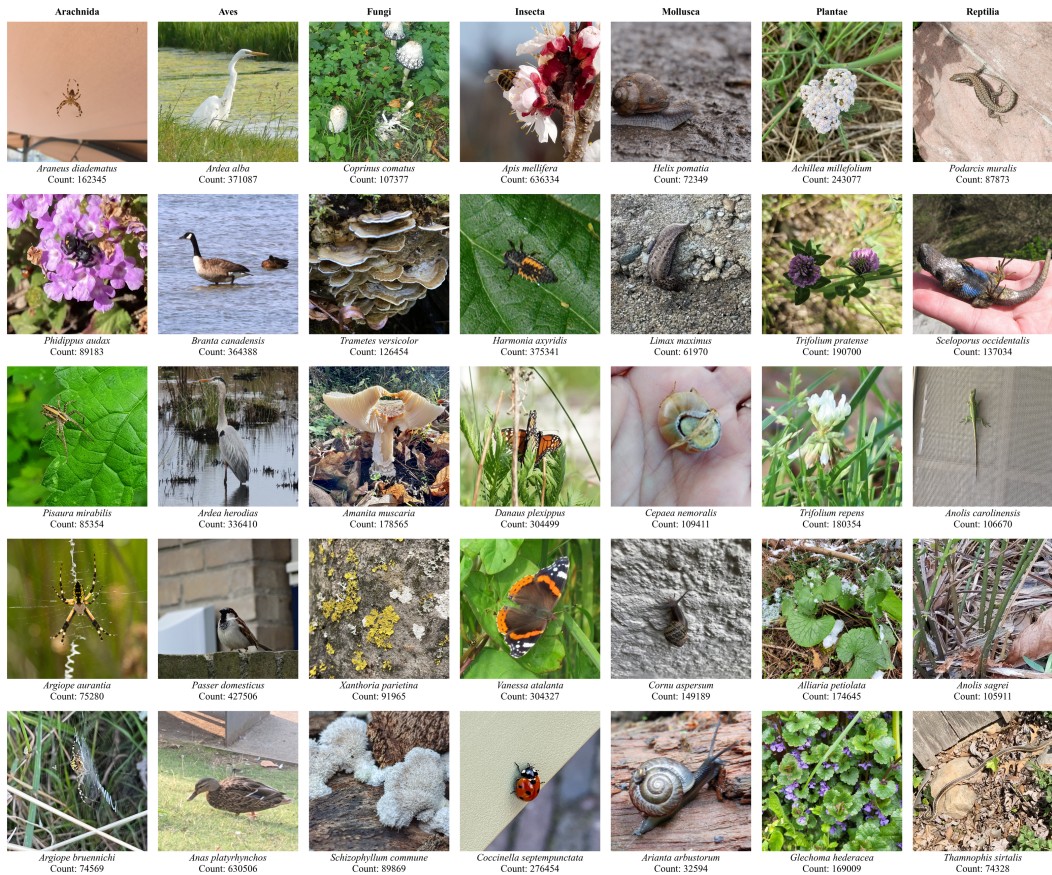

Figure 8: Examples of top five most frequent species and their counts in CRYPTICBIO.

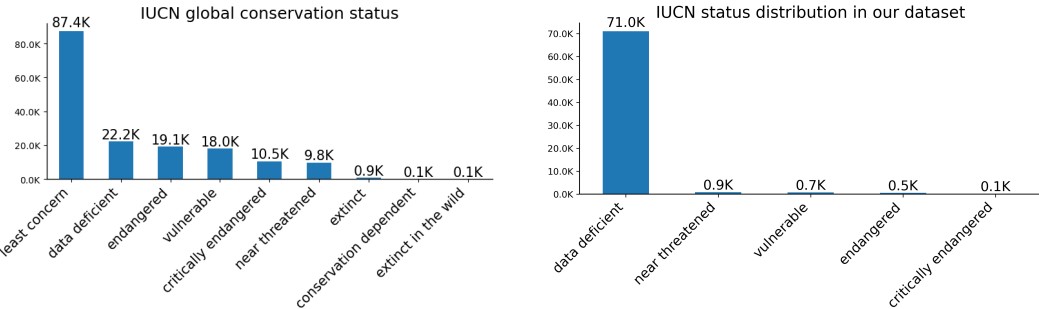

Figure 9: IUCN endangered species distribution: (left) IUCN endangered species distribution, (right) IUCN endangered species distribution in our dataset.

## B.4 Bias in vernacular species terminology across diverse taxa

While we acknowledge the importance of incorporating common species terms alongside scientific (i.e., Latin binomial) nomenclature to enhance model performance [47], the exclusive reliance on English vernacular names risks marginalizing indigenous and non-Western terminologies. Moreover, English speaking cultures may have their regional bias as well. For instance, species *Perisoreus canadensis* is commonly referred to as the *Canada Jay* in Canada, while in the United States is referred to as *Gray Jay* [36]. Currently, datasets like TREEOFLIFE-10M [47] and BIOTROVE [51] include only one version of a species's vernacular name. We believe integrating **multicultural and multilingual common terminology** preserves ecological knowledge and equity, and increases

inclusivity and cultural reach. Thus, we emphasize on enriching species scientific terminology with all common terms from iNaturalist Taxonomy [30] (see dataset details is shown in Section C.3.)

Table 5 shows our dataset and comparable datasets TREEOFLIFE-40M and CRYPTICBIO recorded English vernacular terminology for the widespread flower species *Bellis perennis*. We include English vernacular names in CRYPTICBIO, and provide a pipeline in CRYPTICBIO-CURATE to enrich the dataset with language specific terminology. As illustrated in Figure 10, approximately 30% of species are associated with two or more English vernacular terms, whereas 15% lack any recorded English terminology. It is worth noting that there are also species that have no vernacular names in English, which underlines the importance of preserving indigenous terminology.

Table 5: *Bellis perennis* English vernacular names in existing biodiversity datasets and ours.

| Dataset | Common name |
|---|---|
| TREEOFLIFE-40M | *English daisy* |
| BIOTROVE | *Lawn daisy* |
| CRYPTICBIO | *Common daisy*, *English daisy*, *Lawn daisy* |

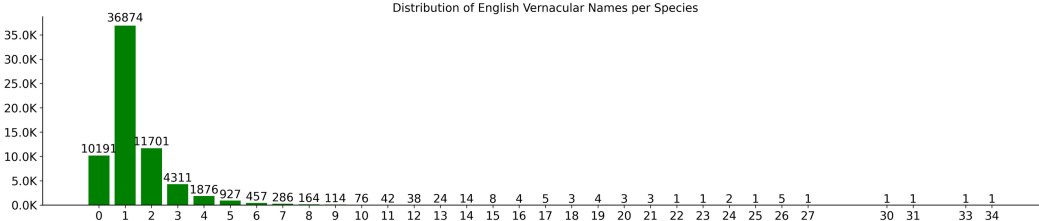

Figure 10: English vernacular names distribution in CRYPTICBIO based on Naturalist Taxonomy [30].

## C  Dataset suite

Table 6 summaries all datasets used for the contruction of CRYPTICBIO, while section C.1-C.4 detail each data source.

Table 6: Dataset suite used in the curation of CRYPTICBIO.

| Dataset | Description |
|---|---|
| GBIF [11] | Occurrence records including species observations with associated metadata such as date, location, and scientific name. Served as the primary source of biodiversity data. |
| GBIF Backbone Taxonomy [45] | Taxonomic reference for resolving scientific names and aligning species-level classifications across datasets. |
| iNaturalist Taxonomy [30] | Cross-referencing vernacular and scientific names, and refining taxonomic granularity particularly for user-contributed observations. |
| iNaturalist "Similar Species" | Cryptic group composed of other species commonly misidentified with a focal species |

### C.1  GBIF

GBIF [11] primarily aggregates research-grade biodiversity data, focusing on species occurrence records derived from scientific sources such as museum collections, academic research, and validated citizen science observations. As a result, the dataset emphasizes verifiable, expert-curated information rather than general public or commercial data.

GBIF uses the Darwin Core standard—a widely adopted vocabulary for sharing biodiversity data. Each GBIF dataset is typically a Darwin Core Archive, structured follows:

- A core CSV file `occurrence.txt` with observation or specimen records. Each row in `occurrence.txt` is one occurrence (a species observation or specimen record).
- Optional extension files: `multimedia.txt`, `identification.txt`.
- A `meta.xml` file describing the structure.
- A `eml.xml` file with metadata about the dataset.

All GBIF occurrence data downloads generated DOI are shown in Table 8. Table 7 summarizes fields used in creation of CONFOUNDINGBIO.

**Occurrences** GBIF's `occurrence.txt` file enumerates 223 fields, however, many fields are often empty. Original identifiers and provenance data files, such as dataset's ID and name (iNaturalist Research-Grade Observations and Observation.org) and original record's unique ID. One of the most important metadata fields is `basisOfRecord`, which tells what kind of occurrence the record is—for example, whether it is a direct human observation, a museum specimen, or machine-generated.

Extensive biological and taxonomic information enumerates full scientific name (usually with authorship), taxonomic level of the record (`taxonRank`), taxonomic hierarchy broken into separate fields (i.e., `kingdom`, `phylum`, `class`, `order`, `superfamily`, `family`, `tribe`, `subtribe`, `subfamily`, `genus`, `subgenus`) and common name (if provided). An important field is `taxonomicStatus` which records the status of the observation (either species, genus, family) is the currently valid/recognized name in taxonomy (marked as `ACCEPTED`) or a synonym and its usage is questionable, incorrectly used. We use only `ACCEPTED` `taxonomicStatus` of `taxonRank` species.

Table 7: GBIF core CSV files `occurrence.txt` and `multimedia.txt` essential field description.

| Field | Description |
|---|---|
| `gbifID` | Unique identifier for occurrence records |
| `scientificName` | Species observation scientific name |
| `taxonRank` | Observation taxonomic level (i.e. species, genus, family) |
| `decimalLatitude`, `decimalLongitude` | Geographic coordinates in decimals |
| `year`, `month`, `day` | Date parts (often included separately too) |
| `type` | The type of media available usually `StillImage`, `Sound`, or `MovingImage`. |
| `identifier` | Direct URL to raw media content (image/audio/video) |
| `license` | Data license (usually CC-BY or CC0) |

The extensive geographic location fields describe where an organism was observed or collected. Core fields detail latitude and longitude coordinates in decimal degrees and radius of uncertainty around the point in meters (e.g. 30, meaning $\pm30$ meters). More locality details include country details and well as free-text description of the place also written in other languages than English.

Apart from `license`, the main field used for legal reuse, there may be detailed access and rights data, `accessRights`, `rightsHolder` giving more contextual info about the data accessibility than the strict license field, however, rarely populated in GBIF records.

Other information less relevant for the modern biodiversity is geological context set of fields (13 fields) in GBIF's, designed to describe the stratigraphic and temporal layers from which a fossil or subfossil specimen was recovered. These fields are especially important for paleontology, stratigraphy, and earth history research and is relevent when an observation is a fossil (i.e., `basisOfRecord` is `FOSSIL_SPECIMEN`).

Data quality fields are critical for assessing whether a record is usable, reliable, or problematic. We use `hasGeospatialIssues` boolean to filter all includes valid geographic coordinates. Another interesting field `iucnRedListCategory` categories taxon conservation status accoridng to International Union for Conservation of Nature (IUCN) Red List [32], although this data is not consistent throughout the records.

**Multimedia** GBIF's `multimedia.txt` file enumerates 15 fields, and can be joined to `occurrence.txt` via `gbifID`. These fields provide access to the media itself (i.e., `identifier`) and its context (i.e., `type`), and specify who created or owns the media, and how it can be used (i.e., `license`).

Table 8: GBIF occurrence download DOIs using in the creation of CRYPTICBIO.

| Group | DOI |
|-------|-----|
| Arachnida (iNaturalist + Observation.org) 03 Apr 2025 [12] | `https://doi.org/10.15468/dl.7sagsw` |
| Aves (iNaturalist) 23 Jan 2025 [14] | `https://doi.org/10.15468/dl.ezf88w` |
| Aves (Observation.org) 23 Jan 2025 [13] | `https://doi.org/10.15468/dl.umgadx` |
| Fungi (iNaturalist + Observation.org) 23 Jan 2025 [15] | `https://doi.org/10.15468/dl.6vb583` |
| Insect (iNaturalist) 23 Jan 2025 [17] | `https://doi.org/10.15468/dl.z7fgt2` |
| Insect (Observation.org) 23 Jan 2025 [16] | `https://doi.org/10.15468/dl.mbmsmm` |
| Mollusca (iNaturalist + Observation.org) 13 Apr 2025 [18] | `https://doi.org/10.15468/dl.eg3pv4` |
| Plantae (iNaturalist) 20 Jan 2025 [19] | `https://doi.org/10.15468/dl.59pyzp` |
| Plantae (Observation.org) 20 Jan 2025 [20] | `https://doi.org/10.15468/dl.pz84ny` |
| Squamata (iNaturalist + Observation.org) 03 Apr 2025 [21] | `https://doi.org/10.15468/dl.kjmm6s` |

## C.2   GBIF Backbone Taxonomy

GBIF Backbone Taxonomy [45] is structured as a Darwin Core Archive, as follows:

- A core TSV (Tab-Separated Values) file `Taxon.tsv` the primary taxonomic information.
- Extension files:
  - `VernacularName.tsv` provides common names (vernacular names) for taxa.
  - `TypeAndSpeciment.tsv` lists information about taxonomic identifications of species.
  - `Description.tsv` contains textual descriptions of taxa, offering additional information such as morphology, behavior, or ecology.
  - `Distribution.tsv` provides geographic and ecological information associated with specific taxa.
  - `Reference.tsv` lists bibliographic references related to the taxa.
  - `Multimedia.tsv` links media resources, such as images or sounds, to taxa.
- A `meta.xml` file describing the structure.
- A `eml.xml` file with metadata about the dataset.

Table 9: GBIF Backbone Taxonomy `Taxon.tsv` essential field description.

| Field | Description |
|-------|-------------|
| `canonicalName` | Unique species scientific name (Latin binomial), lowest taxonomic rank |
| `kingdom` | Highest taxonomic rank (Latin uninomial) |
| `phylum` | Second taxonomic rank (Latin uninomial) |
| `class` | Third taxonomic rank (Latin uninomial) |
| `order` | Forth taxonomic rank (Latin uninomial) |
| `family` | Fifth taxonomic rank (Latin uninomial) |
| `genus` | Sixth taxonomic rank (Latin uninomial) |

GBIF observations enumerate taxonomic hierarchy of 11 levels (`kingdom`, `phylum`, `class`, `order`, `superfamily`, `family`, `tribe`, `subtribe`, `subfamily`, `genus`, `subgenus`) broken into separate

Table 10: Diversity in different taxonomy levels in GBIF Backbone Taxonomy [45] (left) and CRYPTICBIO (right).

| Level | Count | | Level | Count |
|-------|-------|---|-------|-------|
| kingdom | 8 | | kingdom | 3 |
| phylum | 169 | | phylum | 14 |
| class | 519 | | class | 56 |
| order | 1953 | | order | 351 |
| family | 15139 | | family | 2036 |
| genus | 268644 | | genus | 17327 |
| species | 3389404 | | species | 67140 |

Table 11: iNaturalist Taxonomy `taxa.csv` and `VernacularNames-english.csv` essential field description.

| Field | Description |
| --- | --- |
| id | Unique identifier for occurrence records |
| scientificName | Species observation scientific name (i.e. Latin binomial) |
| vernacularName | Common or vernacular name (e.g., "Lawn daisy", "English daisy") |
| language | Language of the vernacular name encoded with ISO 639 standard (e.g., en, es, fr) |

fields, however, many fields are often empty. Instead, we use GBIF's Backbone Taxonomy [45] to enrich observations at `species` taxonomic level with six taxonomic hierarchy levels: `kingdom`, `phylum`, `class`, `order`, `family`, `genus`. Diversity in different taxonomy levels in GBIF Backbone Taxonomy [45] and CRYPTICBIO is shown in Table 10. Unlike the selected six taxonomic hierarchy levels, levels like `superfamily`, `tribe`, `subtribe`, `subfamily`,`subgenus` not consistently recorded in taxonomy [31].

## C.3 iNaturalist Taxonomy

iNaturalist Taxonomy [30] is structured as a Darwin Core Archive, as follows:

- A core CSV file `taxa.csv` the primary taxonomic information.

- Extension files: vernacular names CSV files for each language, encoded as `VernacularNames-[language].csv`; there are 1091 language specific CSV files.

- A `meta.xml` file describing the structure.

- A `eml.xml` file with metadata about the dataset.

We include all vernacular terminology in `VernacularNames-english.csv`. We provide a pipeline in CRYPTICBIO-CURATE to enrich the dataset with language specific terminology.

## C.4 iNaturalist Similar Species

The "Similar Species" feature on iNaturalist is designed to assist annotators in distinguishing between species that are often confused due to their similar appearances. This tool is particularly useful for species which share visual characteristics with other species.

This infomation is derived from two primary sources: (1) errors made by iNaturalist's computer vision model, which learns from millions of community-validated images, and (2) patterns of human misidentifications that are later corrected by other users. By combining these signals, iNaturalist highlights the species that are most frequently mistaken for the focal taxon, presenting them in ranked order on the species page. As a result, the "Similar Species" links represent dynamic, confusion-based relationships that evolve as new observations are added and models are retrained, offering a complementary perspective to formal taxonomic hierarchies by capturing practical field-level identification challenges. An example of such list is shown in Figure 11 for species *Calidris pygmaea*.

This feature is not always visible for all species. Its presence depends on the availability of sufficient observation data and a documented history of misidentifications between the focal species and others. For many less-observed, rare, or underrepresented taxa, the feature may not appear at all. This is because the system relies entirely on community-driven data and automated algorithms that detect patterns in user identifications; it is not manually curated. As a result, even if a species has close look-alikes, the "Similar Species" tab may be absent if those confusions have not been frequently recorded by users. This limitation is especially noticeable for obscure species or those from poorly documented regions, underscoring the importance of consulting external field guides or expert communities when the feature is not available. Because the exact algorithm and weighting used by iNaturalist are not fully disclosed, the "Similar Species" tab remains a black box, limiting interpretability and making it challenging to fully assess coverage and consistency across taxa.

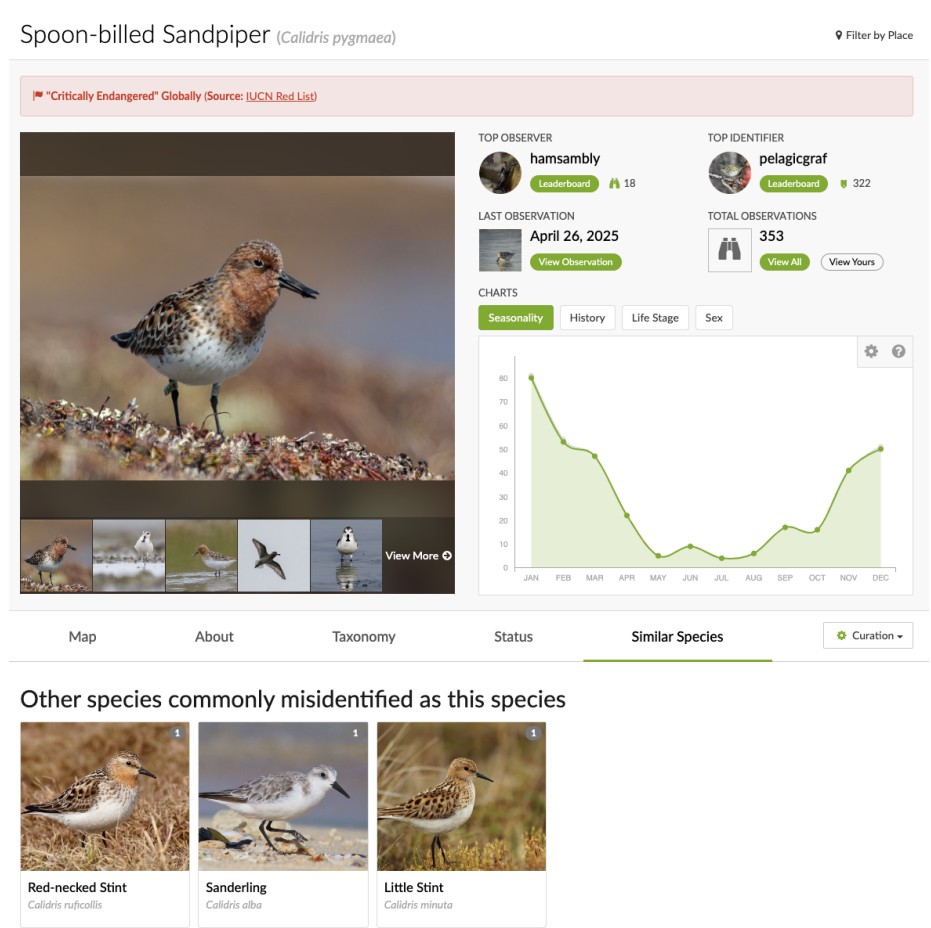

Figure 11: iNaturalist "Similar Species" tab for *Calidris pygmaea*.

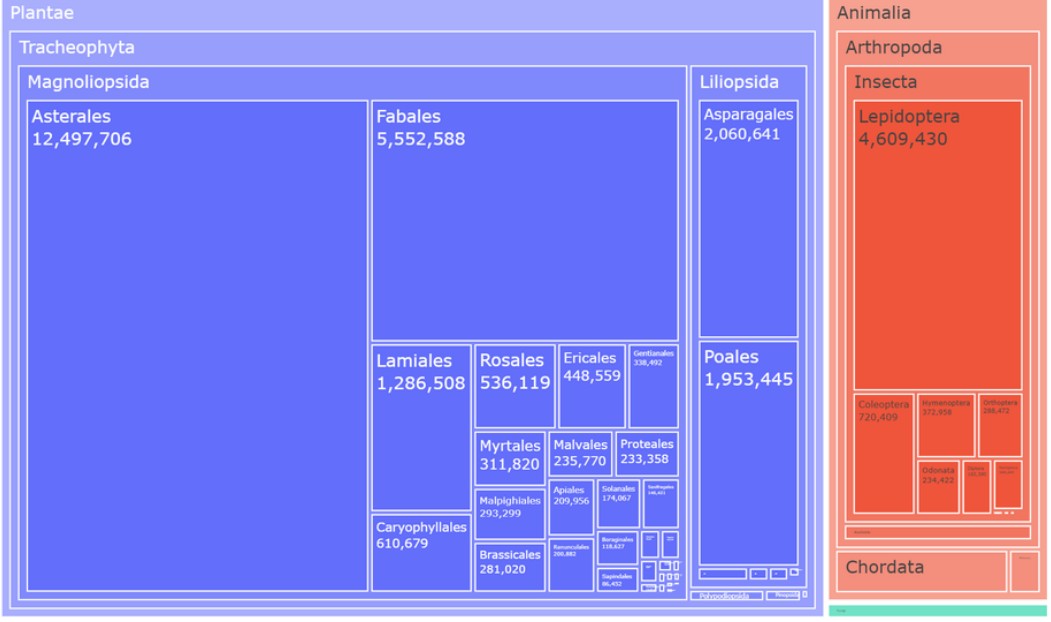

Figure 12: Treemap diagram, starting from *kingdom*. The nested boxes represent *phyla*, *classes*, *orders*, and *families*. Box size represents the relative number of samples in the dataset.

# D CRYPTICBIO dataset

Figure 12 shows the treemap diagram, from *kingdom*, *phyla*, *classes*, *orders*, and *families*, Table 12 shows comparable datasets, and Table 13 shows the overlap of CRYPTICBIO with the public datasets.

Table 12: CRYPTICBIO comparable datasets and benchmarks.

| Dataset | Images | Species | Annotations | Source | Features |
|---|---|---|---|---|---|
| CRYPTICBIO | 166.0M | 71.0K | common (multicultural and multilingual), scientific terms, taxonomic hierarchies, location, date, confounding species groups | GBIF (iNaturalist and Observation.org), GBIF Backbone Taxonomy [45], iNaturalist Taxonomy [30] | multimodal, data-driven cryptic species groups (52K groups) |
| BIOTROVE [51] | 161.9M | 366.6K | common, scientific terms, taxonomic hierarchies | iNaturalist | biased common species terminology annotations |
| TREEOFLIFE-10M [47] | 10.4M | 454.1K | common, scientific terms, taxonomic hierarchies | iNaturalist, Encyclopedia of Life (EOL)[9], BIOSCAN-1M[22] | biased common species terminology annotations |
| TAXABIND-8K [44] | 8.8K | 2.2K | common, scientific term, taxonomic hierarchies, location, environmental features, audio recordings, satellite imagery | iNaturalist, iNat2021[48], Santinel-2[6], WorldClim-2.1[10] | multimodal |
| INATURALIST 2024 [49] | 5M | 10K | scientific terms, location, time | iNaturalist (from 2021–2024) | species curated from iNat2021 [48] |
| AMI-GBIF [33] | 2.5M | 5.3K | scientific terms | GBIF (iNaturalist, Observation.org, Artportalen, Norwegian SOS, Fennoscandia) | manually selected cryptic species group (1 group) |
| BUMBLE BEES [46] (not publicly available) | 89K | 36 | scientific terms | iNaturalist, Bumble Bee Watch [24], BugGuide [2] | manually selected cryptic species group (1 group) |
| TURTLES [1] (not publicly available) | 6.9K | 36 | common, scientific terms | Internet | manually selected cryptic species group (1 group) |
| AMAZON PARROTS [34] | 14K | 35 | scientific terms | iNaturalist, eBird [8], Google Images | manually selected cryptic species group (16 groups) |
| CONFOUNDING SPECIES [4] (not publicly available) | 100 | 10 | scientific term, confounding species pairs | iNaturalist | manually selected cryptic species pairs |
| SQUAMATA LACERTIDAE PODARCIS [42] | 4.0K | 9 | scientific terms | personal collection during field surveys | manually selected cryptic species group (1 group) |
| CHIROPTERA RHINOLOPHIDAE RHINOLOPHUS [3] | 293 | 7 | scientific terms | personal collection during field surveys | manually selected cryptic species group (1 group) |

Table 13: Overlap of CRYPTICBIO species with other datasets.

| Dataset | Species | Overlap | % in CRYPTICBIO | % of CRYPTICBIO | Jaccard |
|---|---|---|---|---|---|
| BIOTROVE [51] | 366.6K | 58.2K | 15.9% | 82.0% | 13.7% |
| TREEOFLIFE-10M [47] | 454.1K | 61K | 13.4% | 85.9% | 12.0% |
| TAXABIND-8K [44] | 2.2K | 2K | 90.9% | 2.8% | 2.7% |
| INATURALIST 2024 [49] | 10K | 7.9K | 79.0% | 11.1% | 10.8% |
| AMI-GBIF [33] | 5.3K | 5.2K | 98.1% | 7.3% | 6.9% |
| AMAZON PARROTS [34] | 35 | 32 | 91.4% | 0.05% | 0.05% |
| SQUAMATA LACERTIDAE PODARCIS [42] | 9 | 8 | 88.9% | 0.01% | 0.01% |

Our dataset leverages real-world confusion patterns to identify perceptually similar species, which we see as a valuable signal of cryptic relationships. However, we fully acknowledge that VLMs could indeed be valuable tools for identifying visual similarities between species by capturing semantic and perceptual relationships at scale. At the same time, we also note that VLMs remain computationally expensive and not free from biases. These models can be influenced by consistent background environments, lighting conditions, or common camera angles, which may not reflect actual visual similarity between species themselves. In future work, we are planning to explore incorporating model-based similarity signals to complement our crowd-sourced approach.

# E    New benchmarks

## E.1    CRYPTICBIO-COMMON benchmark details

We randomly select species from each taxonomic group *Arachnida*, *Aves*, *Fungi*, *Insecta*, *Mollusca*, *Plantae*, and *Reptilia* and corresponding visually confusion group species for each benchmark. Figure 13–19 show examples of cryptic groups in CRYPTICBIO, while Table 14 shows selected species subset distribution. For benchmarking we randomly select 100 images for each species.

Table 14: CRYPTICBIO-COMMON subset distribution.

| Taxon | Selected species | #Associated cryptic species | #Observations |
|---|---|---|---|
| (*Arachnida*) | *Parasteatoda tepidariorum* | 24 | 500425 |
| (*Aves*) | *Passer domesticus* | 25 | 2836119 |
| (*Fungi*) | *Amanita muscaria* | 24 | 333551 |
| (*Insecta*) | *Harmonia axyridis* | 25 | 947893 |
| (*Mollusca*) | *Cornu aspersum* | 25 | 542833 |
| (*Plantae*) | *Bellis perennis* | 24 | 839657 |
| (*Reptilia*) | *Zootoca vivipara* | 19 | 385535 |

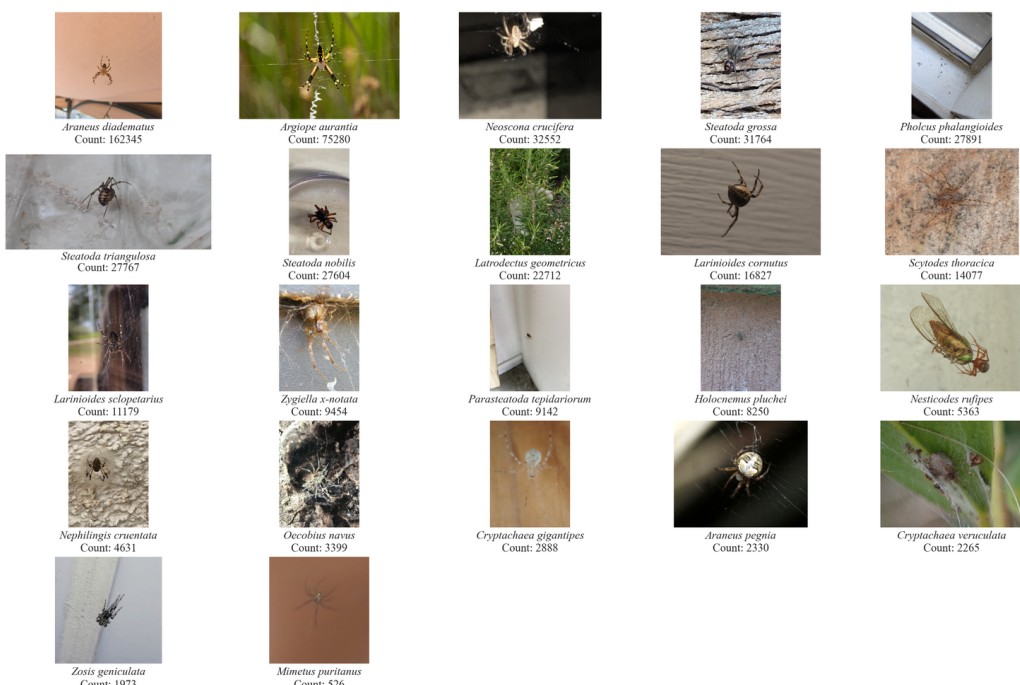

Figure 13: Sample of commonly misidentified of selected species (*Arachnida*) *Parasteatoda tepidariorum*.

**Commonly misidentified as Passer domesticus**

*Passer domesticus*
Count: 432802

*Turdus migratorius*
Count: 300798

*Sturnus vulgaris*
Count: 281847

*Haemorhous mexicanus*
Count: 242829

*Fringilla coelebs*
Count: 205426

*Melospiza melodia*
Count: 204049

*Junco hyemalis*
Count: 178276

*Mimus polyglottos*
Count: 177492

*Zonotrichia leucophrys*
Count: 117751

*Poecile atricapillus*
Count: 117000

*Passer montanus*
Count: 113328

*Zonotrichia albicollis*
Count: 90028

*Spizella passerina*
Count: 88346

*Thryothorus ludovicianus*
Count: 73623

*Prunella modularis*
Count: 61865

*Poecile carolinensis*
Count: 47418

*Zonotrichia atricapilla*
Count: 37279

*Zonotrichia capensis*
Count: 28981

*Chondestes grammacus*
Count: 26258

*Spizelloides arborea*
Count: 23853

*Spizella pusilla*
Count: 20000

*Passer italiae*
Count: 8880

*Passer hispaniolensis*
Count: 7436

*Petronia petronia*
Count: 3550

*Passer melanurus*
Count: 3225

Figure 14: Sample of commonly misidentified of selected species (*Aves*) *Passer domesticus*.

**Commonly misidentified as Amanita muscaria**

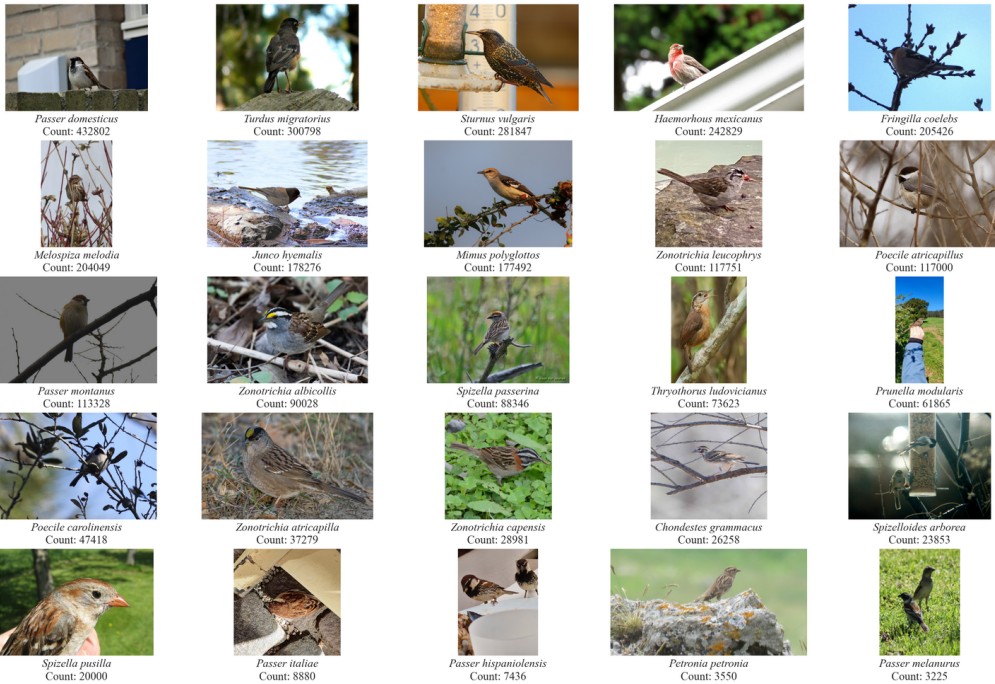

*Amanita muscaria*
Count: 178565

*Amanita rubescens*
Count: 44173

*Leratiomyces ceres*
Count: 26302

*Amanita citrina*
Count: 19514

*Amanita pantherina*
Count: 12759

*Amanita flavoconia*
Count: 9045

*Amanita persicina*
Count: 6118

*Amanita jacksonii*
Count: 5381

*Amanita parcivolvata*
Count: 5264

*Amanita augusta*
Count: 5211

*Amanita flavorubens*
Count: 3988

*Amanita xanthocephala*
Count: 3393

*Amanita pantherinoides*
Count: 2863

*Amanita frostiana*
Count: 1611

*Amanita crenulata*
Count: 1552

*Amanita aprica*
Count: 1502

*Russula sanguinea*
Count: 1340

*Amanita regalis*
Count: 1299

*Amanita caesarea*
Count: 1285

*Amanita basii*
Count: 774

*Amanita velatipes*
Count: 621

*Amanita cokeri*
Count: 601

*Amanita wellsii*
Count: 390

Figure 15: Sample of commonly misidentified of selected species (*Fungi*) *Amanita muscaria*.

**Commonly misidentified as Harmonia axyridis**

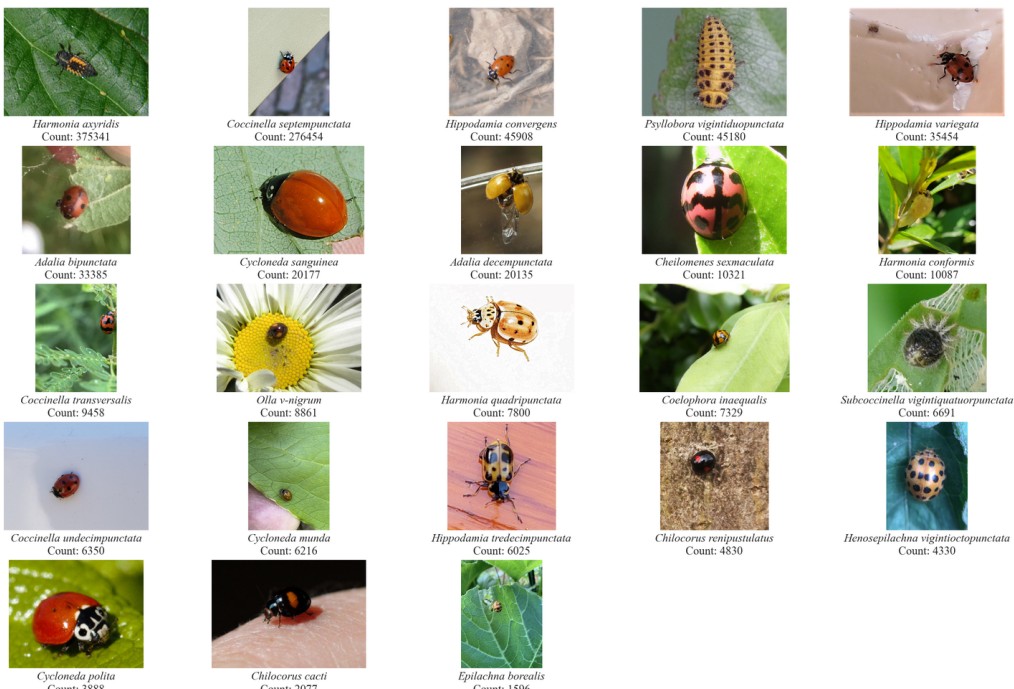

*Harmonia axyridis*
Count: 375341

*Coccinella septempunctata*
Count: 276454

*Hippodamia convergens*
Count: 45908

*Psyllobora vigintiduopunctata*
Count: 45180

*Hippodamia variegata*
Count: 35454

*Adalia bipunctata*
Count: 33385

*Cycloneda sanguinea*
Count: 20177

*Adalia decempunctata*
Count: 20135

*Cheilomenes sexmaculata*
Count: 10321

*Harmonia conformis*
Count: 10087

*Coccinella transversalis*
Count: 9458

*Olla v-nigrum*
Count: 8861

*Harmonia quadripunctata*
Count: 7800

*Coelophora inaequalis*
Count: 7329

*Subcoccinella vigintiquatuorpunctata*
Count: 6691

*Coccinella undecimpunctata*
Count: 6350

*Cycloneda munda*
Count: 6216

*Hippodamia tredecimpunctata*
Count: 6025

*Chilocorus renipustulatus*
Count: 4830

*Henosepilachna vigintioctopunctata*
Count: 4330

*Cycloneda polita*
Count: 3888

*Chilocorus cacti*
Count: 2077

*Epilachna borealis*
Count: 1596

Figure 16: Sample of commonly misidentified of selected species (*Insecta*) *Harmonia axyridis*.

**Commonly misidentified as Cornu aspersum**

*Cornu aspersum*
Count: 149189

*Cepaea nemoralis*
Count: 109411

*Helix pomatia*
Count: 72349

*Arianta arbustorum*
Count: 32594

*Lissachatina fulica*
Count: 25662

*Cepaea hortensis*
Count: 18296

*Hygromia cinctella*
Count: 17581

*Theba pisana*
Count: 17227

*Rumina decollata*
Count: 17083

*Otala lactea*
Count: 15720

*Bradybaena similaris*
Count: 11804

*Eobania vermiculata*
Count: 10247

*Fruticicola fruticum*
Count: 10104

*Helix lucorum*
Count: 5959

*Helicina orbiculata*
Count: 5451

*Monacha cantiana*
Count: 5113

*Zachrysia provisoria*
Count: 4918

*Cantareus apertus*
Count: 2654

*Helminthoglypta tudiculata*
Count: 2501

*Otala punctata*
Count: 2388

*Xerotricha conspurcata*
Count: 2346

*Xeroplexa intersecta*
Count: 1888

*Helminthoglypta nickliniana*
Count: 1750

*Xerarionta stearnsiana*
Count: 588

Figure 17: Sample of commonly misidentified of selected species (*Mollusca*) *Cornu aspersum*.

**Commonly misidentified as Bellis perennis**

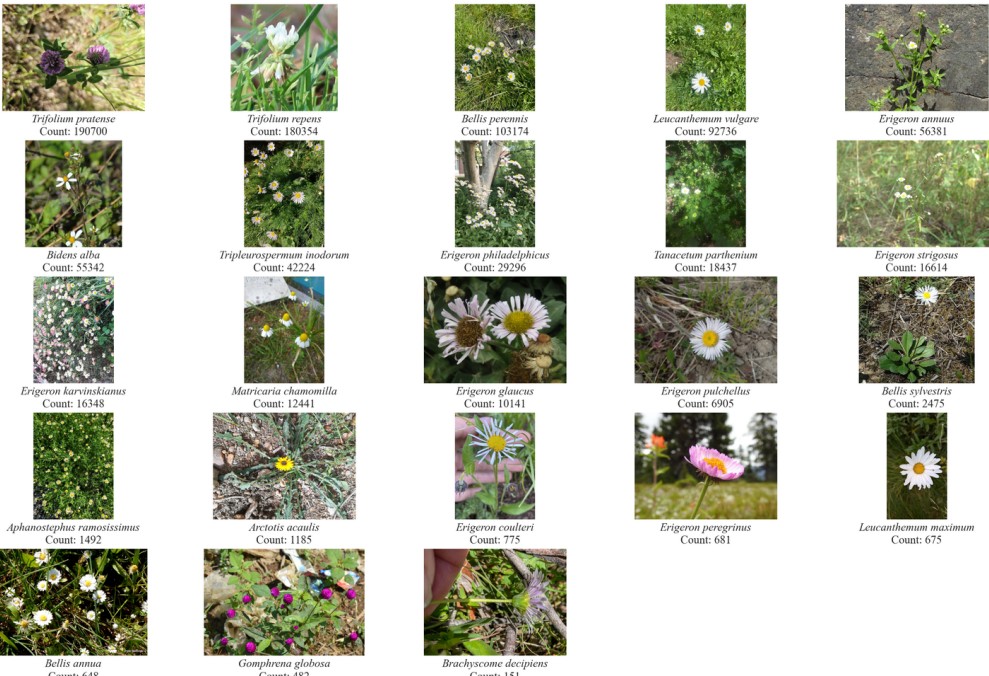

Figure 18: Sample of commonly misidentified of selected species (*Plantae*) *Bellis perennis*.

**Commonly misidentified as Zootoca vivipara**

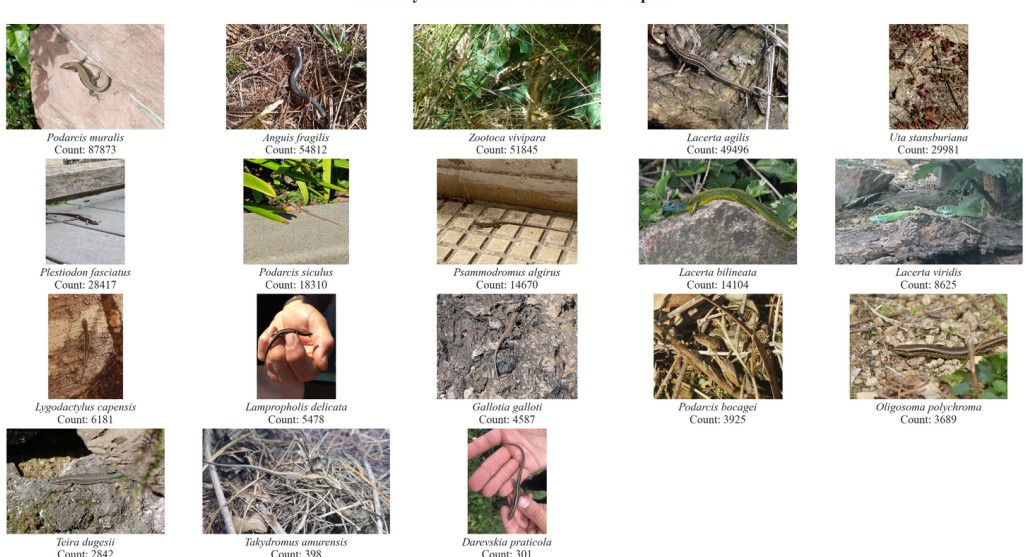

Figure 19: Sample of commonly misidentified of selected species (*Reptilia*) *Zootoca vivipara*.

## E.2 CRYPTICBIO-COMMONUNSEEN benchmark details

We strictly select CRYPTICBIO-COMMON to taxa observed from 01-09-2024 to 01-04-2025. This, we ensure that we evaluate zero-shot learning of established state-of-the-art models using new observations (i.e. images). We randomly select 100 images for each species, spanning n = 133 species (26 species less than CRYPTICBIO-COMMON).

### E.3    CRYPTICBIO-ENDANGERED benchmark details

To highlight the challenges of species identification within conservation-critical contexts, we introduce CRYPTICBIO-ENDANGERED, a curated subset of cryptic species that are listed as threatened or endangered according to the global IUCN Red List [32]. This subset is designed to assess model performance on taxa where misidentification may carry heightened ecological and conservation risks.

We select a species from each taxonomic groups—*Arachnida*, *Aves*, *Fungi*, *Insecta*, *Mollusca*, *Plantae*, and *Reptilia*—each of which contains species that are both visually similar and conservation-relevant. From each group, we randomly select 100 representative samples and their corresponding cryptic species groups. To ensure data quality and sufficient representation for evaluation, we filter out taxa with fewer than 150 recorded observations.

This subset emphasizes the importance of accurate classification for threatened taxa, where even minor identification errors can undermine conservation priorities and downstream ecological analyses. Table 15 further detail the sample characteristics.

Table 15: CRYPTICBIO-ENDANGERED subset distribution.

| Taxon | Selected species | #Associated cryptic species | #Observations |
|---|---|---|---|
| (*Arachnida*) | *Dolomedes plantarius* | 2 | 2352 |
| (*Aves*) | *Calidris ruficollis* | 4 | 129885 |
| (*Fungi*) | *Hygrocybe intermedia* | 3 | 16540 |
| (*Insecta*) | *Petalura gigantea* | 3 | 739 |
| (*Mollusca*) | *Pinna nobilis* | 4 | 2592 |
| (*Plantae*) | *Guaiacum officinale* | 6 | 13981 |
| (*Reptilia*) | *Vipera aspis vivipara* | 15 | 126034 |

### E.4    CRYPTICBIO-INVASIVE benchmark details

To address the increasing ecological risks posed by invasive alien species (IAS), we introduce CRYPTICBIO-INVASIVE, a dedicated benchmark subset focusing on invasive species and their cryptic species selected from the 100 of the World's Worst Invasive Alien Species by Global Invasive Species Database (GISD) [23]. IAS are recognized as a major driver of biodiversity loss, with their occurrences showing exponential growth worldwide [37]. Accurate identification of invasive taxa—particularly those embedded within morphologically cryptic species complexes—is therefore critical for early detection, monitoring, and mitigation efforts.

In constructing this subset, we select 100 representative samples from each cryptic group associated with a selected invasive species. To ensure statistical robustness and adequate representation, we exclude taxa for which fewer than 150 validated observations are available. We select one species for each taxons (i.e. *Aves*, *Fungi*, *Insecta*, *Mollusca*, *Plantae*, excluding *Arachnida* and *Reptilia* as there are not species mentionings in 100 of the World's Worst Invasive Alien Species by Global Invasive Species Database (GISD)).

CRYPTICBIO-INVASIVE highlights the unique challenge of identifying invasive taxa that are visually indistinguishable from native or non-invasive relatives, and serves as a targeted testbed for evaluating model performance in scenarios with direct ecological and policy implications. Table 16 further detail the sample characteristics.

Table 16: CRYPTICBIO-INVASIVE subset distribution.

| Taxonomic group | Selected species | #Associated cryptic group | #Observations |
|---|---|---|---|
| (*Aves*) | *Acridotheres tristis* | 25 | 34689 |
| (*Fungi*) | *Cryphonectria parasitica* | 4 | 672 |
| (*Insecta*) | *Linepithema humile* | 24 | 15178 |
| (*Plantae*) | *Acacia mearnsii* | 25 | 14579 |

## F  EXTENDED RESULTS

Table 17: Model suite used in our benchmarks.

| Modality | Model | | Architecture |
|---|---|---|---|
| Image | **BIOCLIP** [47] | **BC** | ViT-B-16 |
| | **BIOTROVE BIOCLIP** [51] | **BT-B** | ViT-B-16 |
| | **BIOTROVE OPENAI** [51] | **BT-O** | ViT-B-16 |
| | **TAXABIND** [44] | **TB** | ViT-B-16 |
| Geolocation | **TAXABIND** [44] | **L** | GEOCLIP [50] |
| Environment features | **TAXABIND** [44] | **E** | ResNet-Style MLP [5] |

**Experimental details** We benchmark state-of-the-art CLIP-style biodiversity models and the species taxonomic level. We evaluate on Nvidia GeForce RTX 3080 GPU using 32 GB of RAM memory. Table 17 summaries our model suite. We use BIOCLIP [47], BIOTROVE's BIOCLIP ViT-B-16 and OpenAI ViT-B-16 fine-tuned variants [51], and TAXABIND [44] as image-only baseline models.

For multimodal evaluation, we add embeddings obtained from the image encoders to those obtained from TAXABIND location and environmental features encoders, which are then used for zero-shot classification. We collect from WorldClim-2.1[10] environmental features for each observation based on the location metadata, which are then passed through TAXABIND's environmental encoder. We compare performance on scientific, vernacular, and mixed text types, as advised in [47]. We combine image with location and environment embeddings by adding each embedding.

We benchmark on all English available vernacular terminology, and we use species scientific name when vernacular term is missing. Table 18 shows an example of text types. Tables 20–22 summarize performance comparisons across benchmarks on CRYPTICBIO-COMMON, CRYPTICBIO-COMMONUNSEEN, CRYPTICBIO-ENDANGERED, and CRYPTICBIO-COMMON-INVASIVE.

We report zero-shot top-1 accuracy based on cosine similarity. We include a 95% confidence intervals for all reported metrics, calculated using binomial proportion confidence interval method (denoted as $\pm$). Furthermore, we compute an aggregate performance metric, which represents the weighted average accuracy over all classes across the benchmarks. To assess the significance of pairwise performance differences between models, we use McNemar's test (p-value < 0.05).

We deliberately evaluate our approach in a zero-shot setting to assess its generalization capabilities without relying on task-specific fine-tuning. This is an intentional choice aimed at evaluating how well additional contextual information can be leveraged within already existing biodiversity models.

**Results overview** We find combining image and location embeddings to improve performance on zero-shot image classification overall. Models trained with specialist datasets (i.e. BIOTROVE-CLIP and BIOCLIP) perform better. Additionally, mixed scientific and common names yield overall best performance scores, thus, we only report these scores.

We additionally evaluate CRYPTICBIO-COMMONUNSEEN across all BIOTROVE variants, noting that this set comprises taxa observations entirely held out from training.

Table 18: Example of benchmarked text types.

| Text type | Example |
|---|---|
| Scientific | *Bellis perennis* |
| Vernacular | *Common daisy*, *English daisy*, *Lawn daisy* |
| Scientific + Vernacular | *Bellis perennis* commonly known as *Common daisy*, *English daisy*, *Lawn daisy* |

Table 19: CRYPTICBIO-ENDANGERED benchmark Top-1/3/5 accuracy, precision, and recall with 95% confidence intervals. I / L refers to image / location embeddings; BC refers to BIOCLIP; BT-B refers to BIOTROVE-CLIP-BIOCLIP; BT-O refers to BIOTROVE-CLIP-OPENCLIP; TB refers to TAXABIND; MB refers to MULTIMODALBIO.

| Model | Modality | Top-1 | Top-3 | Top-5 | Precision / Recall |
|-------|----------|-------|-------|-------|--------------------|
| BC | I | 49.1±1.63 | 73.0±1.45 | 56.5 ±1.62 | 0.55 / 0.49 ±0.25/0.24 |
| BT-B | I | 58.4 ±1.61 | 77.8 ±1.36 | 61.7 ±1.59 | 0.59 / 0.58 ±0.25/0.25 |
| BT-O | I | 48.9 ±1.63 | 65.7 ±1.55 | 50.9 ±1.63 | 0.48 / 0.49 ±0.23/0.24 |
| TB | I | 52.2 ±1.63 | 75.9 ±1.40 | 59.1 ±1.60 | 0.57 / 0.52 ±0.25/0.24 |
| TB | I+L | 52.3 ±1.63 | 76.1 ±1.39 | 59.3 ±1.60 | 0.57 / 0.52 ±0.25/0.24 |
| MB-BT-B | I+L | **63.2 ±1.57** | **83.3 ±1.22** | **66.6 ±1.54** | 0.65 / 0.63 ±0.27/0.27 |
| MB-BT-O | I+L | 50.5 ±1.63 | 69.1 ±1.51 | 54.6 ±1.63 | 0.49 / 0.51 ±0.23/0.24 |

Table 20: CRYPTICBIO-COMMONUNSEEN benchmark on various models. I / L / E refers to image / location / environmental features embeddings; AR / AV / F / I / M / P / R refers to taxonomic groups *Arachnida / Aves / Fungi / Insecta / Mollusca / Plantae / Reptilia*; MN refers to mixed (scientific + common) text annotations; WA refers to weighted average; BC refers to BIOCLIP; BT-B refers to BIOTROVE-CLIP-BIOCLIP; BT-O refers to BIOTROVE-CLIP-OPENCLIP; TB refers to TAXABIND. Location (L) and environmental features (E) are TAXABIND embeddings.

| Model | Modality | AR-MN | AV-MN | F-MN | I-MN | M-MN | P-MN | R-MN | WA |
|-------|----------|-------|-------|------|------|------|------|------|-----|
| BC | I | 37.6 ±1.58 | 55.1 ±1.62 | 49.7 ±1.63 | 38.5 ±1.59 | 29.5 ±1.49 | 63.7 ±1.57 | 46.0 ±1.63 | 45.76 |
| BT-B | I | 59.1 ±1.61 | 61.4 ±1.59 | 72.7 ±1.45 | 50.5 ±1.63 | 49.2 ±1.63 | **76.4 ±1.39** | 62.0 ±1.58 | 61.66 |
| BT-O | I | 50.8 ±1.63 | 43.4 ±1.62 | 74.5 ±1.42 | 31.7 ±1.52 | 39.7 ±1.60 | 60.0 ±1.60 | 48.4 ±1.63 | 49.84 |
| TB | I | 41.2 ±1.61 | 59.5 ±1.60 | 52.5 ±1.63 | 40.4 ±1.60 | 32.4 ±1.53 | 64.0 ±1.57 | 52.0 ±1.63 | 48.89 |
| TB | I+L | 41.4 ±1.61 | 59.6 ±1.60 | 52.6 ±1.63 | 40.5 ±1.60 | 32.5 ±1.53 | 64.2 ±1.56 | 52.5 ±1.63 | 49.08 |
| BT-B | I+L | **59.5 ±1.60** | **65.1 ±1.56** | 76.3 ±1.39 | **54.9 ±1.62** | **50.2 ±1.63** | 75.0 ±1.41 | **68.7 ±1.51** | **64.29** |
| BT-B | I+E | 25.2 ±1.42 | 30.5 ±1.50 | 37.5 ±1.58 | 14.0 ±1.14 | 21.5 ±1.34 | 26.8 ±1.45 | 27.2 ±1.45 | 26.14 |
| BT-O | I+L | 50.4 ±1.63 | 46.0 ±1.63 | **76.4 ±1.39** | 34.4 ±1.55 | 38.7 ±1.59 | 63.0 ±1.58 | 53.5 ±1.63 | 51.79 |
| BT-O | I+E | 23.1 ±1.38 | 21.9 ±1.35 | 40.7 ±1.60 | 12.4 ±1.08 | 16.4 ±1.21 | 20.4 ±1.32 | 21.1 ±1.33 | 22.33 |

Table 21: CRYPTICBIO-ENDANGERED benchmark on various models. I / L / E refers to image / location / environmental features embeddings; AR / AV / F / I / M / P / R refers to taxonomic groups *Arachnida / Aves / Fungi / Insecta / Mollusca / Plantae / Reptilia*; MN refers to mixed (scientific + common) text annotations; WA refers to weighted average; BC refers to BIOCLIP; BT-B refers to BIOTROVE-CLIP-BIOCLIP; BT-O refers to BIOTROVE-CLIP-OPENCLIP; TB refers to TAXABIND. Location (L) and environmental features (E) are TAXABIND embeddings.

| Model | Modality | AR-MN | AV-MN | F-MN | I-MN | M-MN | P-MN | R-MN | WA |
|-------|----------|-------|-------|------|------|------|------|------|-----|
| BC | I | 53.0 ±1.63 | 49.5 ±1.63 | 60.3 ±1.60 | 74.3 ±1.43 | 33.0 ±1.54 | 60.3 ±1.60 | 27.3 ±1.45 | 51.11 |
| BT-B | I | 48.0 ±1.63 | 43.8 ±1.62 | 50.7 ±1.63 | 48.3 ±1.63 | **44.8 ±1.62** | 51.3 ±1.63 | **34.4 ±1.55** | 45.89 |
| BT-O | I | 45.0 ±1.62 | 26.8 ±1.45 | 64.7 ±1.56 | 46.7 ±1.63 | 36.8 ±1.57 | 47.0 ±1.63 | 21.2 ±1.33 | 41.15 |
| TB | I | **54.0 ±1.63** | 53.3 ±1.63 | 59.7 ±1.60 | 75.0 ±1.41 | 31.8 ±1.52 | 63.2 ±1.57 | 31.5 ±1.52 | **52.62** |
| TB | I+L | **54.0 ±1.63** | 53.0 ±1.63 | 59.3 ±1.60 | 74.7 ±1.42 | 31.5 ±1.52 | **63.3 ±1.57** | 31.6 ±1.52 | 52.48 |
| BT-B | I+L | 46.5 ±1.63 | 47.8 ±1.63 | 45.0 ±1.62 | 44.7 ±1.62 | 44.5 ±1.62 | 55.5 ±1.62 | 32.8 ±1.53 | 45.24 |
| BT-B | I+E | 43.5 ±1.62 | 37.0 ±1.58 | 33.3 ±1.54 | 33.3 ±1.54 | 31.0 ±1.51 | 33.5 ±1.54 | 19.4 ±1.29 | 33.01 |
| BT-O | I+L | 42.0 ±1.61 | 28.0 ±1.47 | **65.0 ±1.56** | 40.0 ±1.60 | 42.0 ±1.61 | 48.2 ±1.63 | 20.5 ±1.32 | 40.81 |
| BT-O | I+E | 40.0 ±1.60 | 26.0 ±1.43 | 35.7 ±1.56 | 33.3 ±1.54 | 31.0 ±1.51 | 32.8 ±1.53 | 11.8 ±1.06 | 30.09 |

Table 22: CRYPTICBIO-INVASIVE benchmark on various models. I / L / E refers to image / location / environmental features embeddings; AV / F / I / P refers to taxonomic groups *Aves / Fungi / Insecta / Plantae*; MN refers to mixed (scientific + common) text annotations; WA refers to weighted average; BC refers to BIOCLIP; BT-B refers to BIOTROVE-CLIP-BIOCLIP; BT-O refers to BIOTROVE-CLIP-OPENCLIP; TB refers to TAXABIND. Location (L) and environmental features (E) are TAXABIND embeddings.

| Model | Modality | AV-MN | F-MN | I-MN | P-MN | WA |
|-------|----------|-------|------|------|------|-----|
| BC | I | 59.91 ±1.60 | 66.75 ±1.54 | 33.61 ±1.54 | 36.17 ±1.57 | 49.11 |
| BT-B | I | 76.05 ±1.39 | 61.25 ±1.59 | 54.72 ±1.63 | 41.42 ±1.61 | 58.36 |
| BT-O | I | 62.15 ±1.58 | 55.75 ±1.62 | 48.11 ±1.63 | 29.77 ±1.49 | 48.95 |
| TB | I | 64.12 ±1.57 | **69.00 ±1.51** | 38.28 ±1.59 | 37.58 ±1.58 | 52.24 |
| TB | I+L | 64.32 ±1.56 | 68.75 ±1.51 | 38.61 ±1.59 | 37.58 ±1.58 | 52.31 |
| BT-B | I+L | **76.77 ±1.38** | 64.75 ±1.56 | **61.06 ±1.59** | **50.33 ±1.63** | **63.23** |
| BT-B | I+E | 18.78 ±1.28 | 57.50 ±1.61 | 21.39 ±1.34 | 22.99 ±1.37 | 30.17 |
| BT-O | I+L | 61.79 ±1.59 | 57.25 ±1.62 | 50.06 ±1.63 | 33.03 ±1.54 | 50.53 |
| BT-O | I+E | 13.18 ±1.10 | 52.75 ±1.63 | 18.11 ±1.26 | 14.21 ±1.14 | 24.56 |

