# OpenReview forum: "CrypticBio: A Large Multimodal Dataset for Visually Confusing Species"
_NeurIPS.cc/2025/Datasets_and_Benchmarks_Track — NeurIPS 2025 Datasets and Benchmarks Track poster_

### Official Review · Reviewer_nxhp · 2025-06-29

**Ethics Flags:** Data privacy, copyright, and consent
**Rating:** 5
**Confidence:** 4

**Summary:**

The paper presents CrypticBio dataset, a dataset of visually confusing species where species are grouped into various cryptic groups. The dataset features 166M images with 52k unique cryptic groups and 67k unique species. The dataset includes rich metadata including timestamp and geographic location where images are captured and multicultural and multilingual species labels. The dataset is primarily curated using the GBIF platform and is carefully filtered to not reveal personal information and only include images with appropriate licenses. The authors benchmark state-of-the-art CLIP-like models on the task of species classification over various subsets of the CrypticBio dataset.

**Additional Feedback:**

This paper presents a technically sound approach and contributes a novel dataset. However, there are minor gaps in the analysis. The dataset requires proper DOI GBIF downloads, and the paper needs to address the questions raised in the Limitations Section before I can raise my score.

**Dataset Code Accessibility:**

Partly

**Dataset Code Comments:**

The provided code includes pipeline to download images from the GBIF platform. However, it does not include the methodology used to group species into specific cryptic groups. It also does not include the code used for benchmarking various models on the cryptic-bio dataset.

**Ethical Comments:**

All GBIF occurrence data downloads require a proper citation using the DOI generated at the time of download (https://www.gbif.org/citation-guidelines). I encourage the authors to provide the DOI information.

**Ethical Considerations:**

Yes, there are ethics concerns that require attention by the authors

**Final Justification:**

The papers presents a novel dataset for cryptic-species classification. The paper benchmarks existing SoTA vision-language models including auxiliary information such as geographic location on this task. The rebuttal has addressed my concerns and I recommend the authors to include the suggested discussions and experiments in the final paper. I am leaning towards acceptance.

**Limitations Weaknesses:**

**Major Weaknesses**

1. I am curious if there are alternative techniques to detect and curate the cryptic species list, such as using VLMs or Large Vision Models (LVMs) in addition to using historical label information. I am saying this because species labels on iNaturalist are prone to human error (although they might be fewer in proportion). Reclassification of an image over time may be a result of human error rather than a species actually being visually confusing to another species. I suppose incorporating VLMs or LVMs can provide an additional check that a species truly belongs to a cryptic group, and rule out the possibility of human error.

2. The flow of Section 2 (CrypticBio Dataset) and Section 3 (Data Curation) is very confusing. In my opinion, content such as in Line 106-112 discussing about the pipeline used to group species into cryptic groups should belong to Section 3. Furthermore, Section 3, especially the curation of cryptic groups needs more details.

3. I would have liked to see the overlap in species classes between the CrypticBio dataset and other datasets as mentioned in Table-11. I further recommend the authors to cite and discuss iNaturalist-2024 [1] dataset and how it compares against the datasets in Table-11.

4. Some important questions I have which have not been discussed in the paper, such as: 1) Is there any correlation between the performance of the models and the size of a cryptic group?; 2) Does the quality of photograph taken play a role in misidentification of species (for example the species being far away from the camera and being almost invisible)?

**Minor Weaknesses**
1. Any reason why the models perform poorly when including environmental variables along with the images?
2. Do the authors have any insights on how current VLMs would perform on the task of cryptic species classification?

References

[1] Vendrow, Edward, et al. "INQUIRE: A natural world text-to-image retrieval benchmark." Advances in Neural Information Processing Systems 37 (2024): 126500-126514.

**Strengths Contributions:**

1. A careful data curation pipeline is developed to group species into visually confusing (cryptic) classes. The authors use historically changing label information of images from the iNaturalist platform to generate the list of cryptic classes.
2. I believe the CrypticBio dataset will be valuable to the vision community, as it will aid in the development of robust methods for visually confusing image classification.
3. The writing is good in general and extensive qualitative and quantitative results are provided in the paper.

---

> ### Author Rebuttal · Authors · 2025-07-26
>
> We sincerely thank the reviewer for the detailed and thoughtful review. We greatly appreciate your recognition of the technical soundness of our work and the value of the CRYPTICBIO dataset to the vision community. Your comments on the careful curation of cryptic species groups, the relevance of our dataset for robust classification tasks, and the quality of writing and analysis are deeply encouraging. We are glad to see the potential impact of CRYPTICBIO acknowledged, and your support reinforces our motivation to advance research in fine-grained and visually confusing species classification.
>
> **"...alternative techniques to detect and curate the cryptic species list ... VLMs or Large Vision Models (LVMs) in addition to using historical label information..." and "Do the authors have any insights on how current VLMs would perform on the task of cryptic species classification?"**
>
> We thank the reviewer for raising this important point. Our work already employs VLMs (domain-adapted versions of CLIP i.e. BioCLIP, BioTrove, and TaxaBind) as core models for evaluating cryptic species identification; and show that image-only inputs often fall short for fine-grained, visually similar species (see Results section lines 209-222). We agree that historical reclassifications may reflect annotation errors in some cases, but we argue that many such reclassifications, particularly those confirmed by experienced iNaturalist contributors, capture genuine visual ambiguity between species. It is worth noting that iNaturalist already uses a large-scale computer vision model to suggest species labels during upload, trained on millions of images (it has been using machine learning since 2017 to provide computer vision suggestions on the iNaturalist website and mobile apps). As a result, many initial labels are close to what the model predicts, and disagreements often emerge in difficult or ambiguous cases. Our current approach leverages these real-world confusion patterns to identify perceptually similar species, which we see as a valuable signal of cryptic relationships. However, we fully acknowledge that VLMs could indeed be valuable tools for identifying visual similarities between species by capturing semantic and perceptual relationships at scale. At the same time, we also note that VLMs remain computationally expensive and not free from biases. These models can be influenced by consistent background environments, lighting conditions, or common camera angles, which may not reflect actual visual similarity between species themselves. In future work, we are planning to explore incorporating model-based similarity signals to complement our crowd-sourced approach.
>
> **"... content such as in Line 106-112 discussing about the pipeline used to group species into cryptic groups should belong to Section 3. Furthermore, Section 3, especially the curation of cryptic groups needs more details."**
>
> Thank you for this helpful observation. We are happy to reorganise the contents in Section 2 and 3 and further extent Section 3 with web-scraping details. We will restructure these sections to clearly separate the high-level overview of the dataset (Section 2) from the detailed description of the cryptic group curation process (Section 3). We will further extend the main paper text with web scraping details, mentioned also in the response to reviewer **jqGY**.
>
> **"...overlap in species classes between the CrypticBio dataset and other datasets as mentioned in Table-11 ... cite and discuss iNaturalist-2024 [1] dataset and how it compares against the datasets in Table-11."**
>
> We thank the reviewer for this thoughtful suggestion. We agree that analysing the overlap between CRYPTICBIO and existing datasets, particularly those in Table 11, adds valuable context. We compute the overlap and plan to include it in the revised version.
>
> |Dataset|Species|Overlap|% in CB|% of CB|Jaccard|
> |-------|------:|------:|------:|------:|-------:|
> |BioTrove|366.6K|58.2K|15.9%|82.0%|13.7%|
> |TreeOfLife-10M|454.1K|61K|13.4%|85.9%|12.0%|
> |TaxaBind-8K|2.2K|2K|90.9%|2.8%|2.7%|
> |Amazon Parrots|35|32|91.4%|0.05%|0.05%|
> |Squamata|9|8|88.9%|0.01%|0.01%|
>
>
> Furthermore, we thank the reviewer for the thoughtful suggestion and for pointing us to the iNaturalist 2024 (iNat24) dataset. iNat24 contains only 4.8M images of 9K species, denoted by the authors "a subset of iNaturalist". Compared to CRYPTICBIO, iNat24 spans more kingdoms (including mammals, fish, and amphibians), but includes significantly fewer species-level classes, especially within densely populated subsets such as birds, plants and insects. In contrast, CRYPTICBIO is designed to densely sample visually cryptic species, making it particularly suitable for fine-grained classification and zero-shot evaluation.
> Additionally, while iNat24 includes only one English vernacular name per species, CRYPTICBIO explicitly supports linguistic variation across regions ("ladybug" vs. "ladybird"). This broader coverage is critical for training and evaluating AI systems intended for global deployment. We plan to add the iNat24 dataset to the related work section. Additionally, as pointed out by reviewer ** jqGY**, we will also expand our related datasets with AMI dataset.
>
> **"Is there any correlation between the performance of the models and the size of a cryptic group?"**
>
> We thank the reviewer for raising this important question. There is a positive correlation between the size of a cryptic group (number of species in a benchmark) and the average performance across all models. The p-values ~0.05 (Spearman) and ~0.07 (Pearson) just above conventional thresholds for statistical significance suggest larger cryptic groups are associated with better model performance. Its worth noting species in larger groups tend to be more common (i.e. CRYPTICBIO-COMMON and CRYPTICBIO-COMMONUNSEEN) and overrepresented in public biodiversity datasets like iNaturalist and Observation.org, resulting in stronger image-text associations even in zero-shot. Smaller cryptic groups often consist of rare or underrepresented species, which pose a greater challenge due to fewer images. Future benchmarks should consider stratifying evaluation by group size and representation level.
>
> **"quality of photograph taken play a role in misidentification of species (for example the species being far away from the camera and being almost invisible)"**
>
> This is a well raised issue. We agree that image quality, including factors like distance to subject, focus, lighting, and occlusion, can significantly impact both human and model species identification accuracy. In our current dataset curation, we did not explicitly filter or stratify images based on image quality. We would like to note that neither of the data sources we use (i.e. GBIF's curated iNaturalist and Observation.org) do not include any form of image quality assessment or filtering. As a result, our dataset may contain a range of photographic conditions, from high-quality close-ups to more challenging, lower-quality images. Furthermore, models employed in evaluation have also been trained on images regardless of the quality. However, this is a valuable direction for future work, which we already address using traditional and AI-based image quality assessment measures. Misidentifications caused by poor photographic conditions may introduce noise that differs from true cryptic confusion due to visual similarity between species.
>
> **"Any reason why the models perform poorly when including environmental variables along with the images?"**
>
> We thank the reviewer for drawing attention to the environmental variables modality. We observe that incorporating environmental variables alongside images results in a significant drop in zero-shot accuracy. We believe this is due to the limited discriminative value of environmental features for fine-grained classification, particularly within cryptic species groups that often share similar habitats. Additionally, environmental embeddings may be coarse, noisy, or misaligned with the image modality, which can dilute visual signals in a shared embedding space. This effect is especially pronounced in zero-shot settings, where model robustness is sensitive to modality noise and fusion quality. We are exploring improved fusion strategies and adaptive weighting of environmental inputs for future work.
>
> **"All GBIF occurrence data downloads require a proper citation"**
>
> We thank the reviewer for this important feedback. We will include the corresponding GBIF download DOI in the revised version. This ensures that the dataset used in our analyses is fully traceable and that proper credit is given to the original data providers.
>
> Below we show the generated DOI:
>
> Mollusca
> (iNat + Obs.org) 13 Apr 25 https://doi.org/10.15468/dl.eg3pv4
>
> Squamata
> (iNat + Obs.org) 03 Apr 25 https://doi.org/10.15468/dl.kjmm6s
>
> Arachnida
> (iNat + Obs.org) 03 April 25 https://doi.org/10.15468/dl.7sagsw
>
> Aves
> (iNat) 23 Jan 25 https://doi.org/10.15468/dl.ezf88w
> (Obs.org) 23 Jan 25 https://doi.org/10.15468/dl.umgadx
>
> Fungi (iNat + Obs.org) 23 Jan 25 https://doi.org/10.15468/dl.6vb583
>
> Insect
> (iNat) 23 Jan 25 https://doi.org/10.15468/dl.z7fgt2
> (Obs.org) 23 Jan 25 https://doi.org/10.15468/dl.mbmsmm
>
> Plantae
> (iNat): 20 Jan 25 https://doi.org/10.15468/dl.59pyzp
> (Obs.org) 20 Jan 25 https://doi.org/10.15468/dl.pz84ny
>
> **"...not include the methodology used to group species into specific cryptic groups. It also does not include the code used for benchmarking various models on the cryptic-bio dataset."**
>
> As noted in our response to reviewer **jqGY**, the groups were constructed by scraping the “Similar Species” tab on iNaturalist using GBIF-derived species lists. We will release the web scraper and added comprehensive documentation in our repository, along with benchmarking scripts to fully reproduce our results.

---

> > ### Comment · Reviewer_nxhp · 2025-08-03
> > **Solid Paper with a Novel Dataset Contribution**
> >
> > I sincerely thank the authors for addressing the comments I raised previously. The rebuttal has addressed most of my concerns. I strongly suggest the discussions on the topics I raised: 1) "alternative methods for cryptic-species curation"; 2) "photograph quality"; 3) "Size of cryptic-groups vs performance" be included in the paper under an appropriate sections. Given these additions, I have updated my score and have no further comments.

---

### Official Review · Reviewer_EwWU · 2025-06-30

**Rating:** 5
**Confidence:** 4

**Summary:**

The proposed dataset is comprised of species from the iNaturalist dataset which were commonly misclassified by contributors.  In this way, CrypticBio uses crowd-sourced, real-world examples of which species are commonly mistaken for other similar species.  The dataset targets a clear gap in the literature for cryptic species and will enable further research on fine-grained classification of species.   The authors also present results of various zero shot learning models on the different datasets and splits created; and also investigates the impact of additional features (location and environmental) on the classification performance.

**Dataset Code Accessibility:**

Partly

**Dataset Code Comments:**

Dataset is accessible and clearly documented through huggingface. The github repo contains the scripts necessary to reproduce the data processing steps as well as scripts for the benchmarks in the paper.  However the README for the code is lacking – it does not contain steps or instructions to recreate the results presented in the paper.  There should be a basic quick start guide with the necessary steps to install packages and run the scripts provided.

**Ethical Comments:**

The authors have discussed issues related to geoprivacy, sensitive content and responsible use of the data.  The dataset aims to aid in conservation and environmental monitoring.

**Ethical Considerations:**

No, there are no or only very minor ethics concerns

**Final Justification:**

I thank the authors for addressing the limitations raised in the review process.  If the authors make the changes as they have indicated, then I recommend the paper for acceptance.  The work presented proposes an interesting new variation of the iNaturalist dataset which will enable fine-grained recognition of cryptic species.

**Limitations Weaknesses:**

-The definition of a cryptic group vs species should be explained more clearly: what does it mean that there are 52K unique cryptic groups, but that the dataset contains 67K species?

-Review the use of the term “biodiversity” – in line 23, it is used in the context of a “real-world-ready biodiversity AI model”, which would assess the baseline biodiversity of an ecosystem, whereas from the context here it seems the authors are referring instead to models which are capable of real-world-ready fine-grained species classification.  This should also be corrected in the title: should it be CrypticBio: A Large Multimodal Dataset for Visually Confusing Species?

-Although the data is essentially a re-processed version of existing datasets, it still constitutes a useful contribution that significantly builds on the original data.

**Strengths Contributions:**

-The dataset contains 166 million images, making it a large scale classification dataset which targets a gap for cryptic species.

-The dataset includes metadata including location, which is relevant for recognition of many species, as well as multiple splits for specific tasks e.g. classification of endangered species.  The species descriptions map the scientific names and vernacular names to the classes.

---

> ### Author Rebuttal · Authors · 2025-07-25
>
> We sincerely thank the reviewer for the thoughtful and constructive feedback. We are grateful for the recognition of CRYPTICBIO''s strengths; especially the value of using real-world, crowd-sourced confusion patterns and the scale and richness of the dataset. Your comments affirm our aim to address a key gap in fine-grained biodiversity research, and we truly appreciate your support.
>
> Below, we address the limitations raised:
>
> **"The definition of a cryptic group vs species should be explained more clearly: what does it mean that there are 52K unique cryptic groups, but that the dataset contains 67K species?"**
>
> We appreciate the reviewer highlighting the need for clarification. In CRYPTICBIO, a cryptic group represents a cluster of species that are visually similar and commonly confused by annotators. These groups are derived from confusion patterns in the iNaturalist dataset and may contain one or more distinct species. Thus, while the dataset contains 67K unique species, these are organized into 52K cryptic groups based on visual confusion and misclassification patterns. Importantly, these groups vary in size and composition; for example, species A may be grouped with B and C due to frequent confusion among them, while species B might only be confused with C, forming a smaller group, suggesting that the groups may not be exclusive, i.e. species are not assigned to exactly one group. We will clarify this distinction in the revised version of the paper.
>
> **"Review the use of the term “biodiversity” – in line 23, it is used in the context of a “real-world-ready biodiversity AI model”, which would assess the baseline biodiversity of an ecosystem, whereas from the context here it seems the authors are referring instead to models which are capable of real-world-ready fine-grained species classification. This should also be corrected in the title: should it be CrypticBio: A Large Multimodal Dataset for Visually Confusing Species?"**
>
> We agree that our use of the term “biodiversity” may lead to ambiguity. Our intention was to emphasize models capable of real-world fine-grained species recognition, not to directly quantify ecosystem biodiversity. We will revise the phrasing in line 23 and the title to more accurately reflect the dataset's focus. The suggested revision to the title, “CRYPTICBIO: A Large Multimodal Dataset for Visually Confusing Species”, aligns well with the dataset’s scope and we are happy to adopt it.
>
> **"Although the data is essentially a re-processed version of existing datasets, it still constitutes a useful contribution that significantly builds on the original data."**
>
> We thank the reviewer for this insightful observation and for recognizing the innovation in our approach. CRYPTICBIO directly supports progress in fine-grained species recognition under real-world conditions, addressing a critical gap not covered by existing datasets. Our aim was to transform raw observational data into a targeted benchmark for cryptic species classification, a crucial and underexplored challenge in biodiversity AI.
>
> **"... the README for the code is lacking – it does not contain steps or instructions to recreate the results presented in the paper. There should be a basic quick start guide with the necessary steps to install packages and run the scripts provided."**
>
> As pointed out, we will update the README in out GitHub repository to include: 1) a quick-start guide; 2) environment setup instructions; 3) clear steps for reproducing the dataset and benchmarks from the paper. We believe these additions will improve accessibility and reproducibility for the community.

---

### Official Review · Reviewer_jqGY · 2025-07-02

**Rating:** 5
**Confidence:** 4

**Summary:**

The paper introduces CrypticBio, a large-scale image dataset of cryptic species, and four benchmarks based upon that dataset. The dataset is curated from citizen science data and limited to species that are commonly misidentified due to their visual similarity. Several existing biological foundation models are evaluated on the four benchmarks, with geographic locations and environmental variables added as additional predictors.

**Additional Feedback:**

- The AMI dataset [1] is another relevant dataset, containing lots of cryptic species
- Grammar and spelling should be improved (e.g. lines 154-157, “ENGENDERED”)

[1] https://www.ecva.net/papers/eccv_2024/papers_ECCV/papers/05373.pdf

**Dataset Code Accessibility:**

Yes

**Dataset Code Comments:**

The dataset is easily accessible on HuggingFace [1] and Python code is available on GitHub [2] to load and curate the dataset.

[1] https://huggingface.co/datasets/gmanolache/CrypticBio
[2] https://github.com/georgianagmanolache/crypticbio

**Ethical Considerations:**

No, there are no or only very minor ethics concerns

**Final Justification:**

The rebuttal addressed my main concerns, i.e. alternative metrics and the question of reproducibility. I have no further comments.

**Limitations Weaknesses:**

## Evaluation
Although not critical, the main limitation in my mind is the evaluation. First, there are metrics beyond weighted top-1 accuracy that I think would be interesting. For example, top-k accuracy for k > 1 or mean reciprocal rank might be more relevant for human-in-the-loop systems. Furthermore, computing weighted accuracy over all classes (assuming weighted by class prevalence) might overemphasize performance on more frequent species. Adding an ablation that evaluates macro-averaged accuracy or precision / recall / F1 vs. the number of samples might be interesting.

## Reproducibility
Another potential limitation is the reproducibility of the dataset curation workflow. The workflow seems to depend on scraping the iNaturalist webpage, which in contrast to an API might change over time. Furthermore, I could not find any information on exactly how the species in the “Similar species” tab are collected. It is also unclear to me whether iNaturalist technically permits scraping the webpage in this way.

**Strengths Contributions:**

What makes this dataset highly interesting is that it is purposefully curated to contain cryptic, i.e. highly visually confusing species. While there exist lots of datasets that contain some cryptic species, to the best of my knowledge, this is the first dataset that exclusively focuses on such species. This makes this dataset novel and highly relevant for advancing fine-grained recognition and environmental priors. The way the cryptic species groups are mined based on citizen science labels is innovative and seems more scalable (though potentially more noisy) than expert knowledge. The evaluation protocol seems sound and I very much appreciate the inclusion of confidence intervals.

---

> ### Author Rebuttal · Authors · 2025-07-25
>
> We sincerely thank the reviewer for the thoughtful and encouraging evaluation and constructive feedback. We are especially grateful for the recognition of CRYPTICBIO’s novelty and impact, and we are pleased that our contributions resonated with the reviewer. In particular, we appreciate the recognition that “this is the first dataset that exclusively focuses on [cryptic species]”, and that it is “novel and highly relevant for advancing fine-grained recognition and environmental priors.” We also thank the reviewer for describing our mining approach as “innovative and more scalable than expert knowledge,” even while acknowledging potential noise. We believe that leveraging citizen science data in this way aligns with practical, real-world applications of biodiversity AI, and we are encouraged by the reviewer's positive assessment of this strategy. We are equally grateful for the endorsement of our evaluation setup.
>
> **"Although not critical, the main limitation in my mind is the evaluation. First, there are metrics beyond weighted top-1 accuracy that I think would be interesting. For example, top-k accuracy for k > 1 or mean reciprocal rank might be more relevant for human-in-the-loop systems. Furthermore, computing weighted accuracy over all classes (assuming weighted by class prevalence) might overemphasize performance on more frequent species. Adding an ablation that evaluates macro-averaged accuracy or precision / recall / F1 vs. the number of samples might be interesting."**
>
> We thank the reviewer for suggesting additional evaluation metrics. For completeness, we refer the reviewer to Appendix E (lines 532-560), where we provide detailed per-benchmark scores and full metric breakdowns. We clarify that the dataset used in our benchmark is balanced by design, with an equal number of samples per class (n=100). As a result, macro- and weighted-average metrics are effectively equivalent and are not affected by class imbalance. In response, we aim to included Top-1, Top-3, and Top-5 accuracy in our benchmarking suite which due to space constraints were not included in the main paper initally. We agree that such metrics are particularly appropriate for classification tasks under class ambiguity, where multiple plausible predictions may exist. Furthermore, we acknowledge that mean reciprocal rank (MRR) could be valuable in future extensions of this work, particularly for ranked retrieval systems or human-in-the-loop applications.
>
> Below, we present metrics for invasive cryptic species, and plan to include all suggested metrics in the revised version of the paper.
>
> | Model                             |           | Top1 (95% CI)             | Top1 Margin | Top3 (95% CI)             | Top3 Margin | Top5 (95% CI)             | Top5 Margin | Precision (95% CI)        | Precision Margin | Recall (95% CI)           | Recall Margin |
> |----------------------------------|------------------|----------------------------|-------------|----------------------------|-------------|----------------------------|-------------|----------------------------|------------------|----------------------------|----------------|
> | BioCLIP                          | IxT      | 49.11 [47.48, 50.74]       | ±1.63       | 73.01 [71.53, 74.43]       | ±1.45       | 56.53 [54.90, 58.14]       | ±1.62       | 0.55 [0.36, 0.86]          | ±0.25            | 0.49 [0.32, 0.79]          | ±0.24          |
> | BioTroveCLIP-BioCLIP             | IxT      | 58.36 [56.74, 59.96]       | ±1.61       | 77.76 [76.36, 79.08]       | ±1.36       | 61.74 [60.15, 63.32]       | ±1.59       | 0.59 [0.38, 0.89]          | ±0.25            | 0.58 [0.38, 0.89]          | ±0.25          |
> | BioTroveCLIP-OpenCLIP            | IxT      | 48.95 [47.31, 50.58]       | ±1.63       | 65.73 [64.16, 67.26]       | ±1.55       | 50.89 [49.26, 52.52]       | ±1.63       | 0.48 [0.30, 0.75]          | ±0.23            | 0.49 [0.32, 0.79]          | ±0.24          |
> | TaxaBind                         | IxT      | 52.24 [50.62, 53.88]       | ±1.63       | 75.92 [74.49, 77.29]       | ±1.40       | 59.13 [57.52, 60.73]       | ±1.60       | 0.57 [0.38, 0.89]          | ±0.25            | 0.52 [0.34, 0.82]          | ±0.24          |
> | TaxaBind                         | I+LxT   | 52.31 [50.67, 53.93]       | ±1.63       | 76.05 [74.63, 77.42]       | ±1.39       | 59.25 [57.64, 60.84]       | ±1.60       | 0.57 [0.38, 0.89]          | ±0.25            | 0.52 [0.34, 0.82]          | ±0.24          |
> | MultimodalBio-BioTrove-BioCLIP   | I+LxT    | 63.23 [61.63, 64.78]       | ±1.57       | 83.30 [82.05, 84.49]       | ±1.22       | 66.59 [65.03, 68.11]       | ±1.54       | 0.65 [0.43, 0.96]          | ±0.27            | 0.63 [0.43, 0.96]          | ±0.27          |
> | MultimodalBio-BioTrove-OpenCLIP  | I+LxT    | 50.53 [48.89, 52.16]       | ±1.63       | 69.05 [67.53, 70.54]       | ±1.51       | 54.61 [52.98, 56.23]       | ±1.63       | 0.49 [0.30, 0.75]          | ±0.23            | 0.51 [0.32, 0.79]          | ±0.24          |
>
> (I = image, T = text, L = location)
>
> **"Another potential limitation is the reproducibility of the dataset curation workflow. The workflow seems to depend on scraping the iNaturalist webpage, which in contrast to an API might change over time. Furthermore, I could not find any information on exactly how the species in the “Similar species” tab are collected. It is also unclear to me whether iNaturalist technically permits scraping the webpage in this way."**
>
> We thank the reviewer for highlighting this important point. We acknowledge that our current workflow relies on scraping the iNaturalist website to extract information from the “Similar Species” tab. This approach was necessary because the species-level similarity data we rely on is not available through the public API or other official channels. However, we reviewed and respected the site’s robots.txt file (publicly available) to ensure that our scraping did not access any disallowed or restricted endpoints. Specifically, we confirm that the taxon pages used to extract “Similar Species” information are not covered by any exclusion rules. While we recognise the limitations this introduces, we will release our scraping code and the associated metadata in our repository, enabling others to reproduce our results under the same conditions. In the current version, we implement rate limiting to introduce delays between requests of 20 seconds, avoiding any undue load on iNaturalist’s servers. We strictly limit our scope to the publicly accessible “Similar Species” tab on species pages, avoiding any user-specific or login-required content. We are actively exploring API-based or officially supported alternatives for future versions.
>
> We plan to extend with detailed web scarping information in the revised version of the paper in Section 3 as well as Appendix B.4.
>
> **"The AMI dataset is another relevant dataset, containing lots of cryptic species."**
>
> We thank the reviewer for recommending discussion of the AMI dataset. Unlike our CrypticBio dataset, AMI is single-modality (image only) and focuses primarily on moth species. In contrast, CrypticBio integrates multimodal context (location, date, taxonomy) and covers thousands of cryptic species across multiple kingdoms. These differences underscore how CrypticBio complements AMI and enables broader biodiversity recognition modelling. We plan to add the AMI dataset to the related work section. Additionally, as pointed out by reviewer **nxhp**, we will also expand our related datasets with iNaturalist2024 (iNat24).
>
> We thank the reviewer for carefully pointing out the grammar and spelling issues, particularly the typo in lines 154–157 (“ENGENDERED”). We have reviewed the manuscript thoroughly and corrected these errors in the revised version.

---

### Official Review · Reviewer_szBE · 2025-07-03

**Rating:** 5
**Confidence:** 4

**Summary:**

This paper introduces CRYPTICBIO, an exceptionally large and novel multimodal dataset focused on the challenging task of identifying visually confusing, or "cryptic," species. Comprising 166 million images across 67,000 species, it is curated from real-world misidentifications on citizen science platforms. The dataset is uniquely enriched with spatiotemporal metadata (location, date) and multilingual vernacular names. The authors also provide four new benchmarks and demonstrate that incorporating geographical context significantly improves the accuracy of state-of-the-art models in identifying these difficult species.

**Dataset Code Accessibility:**

Yes

**Dataset Code Comments:**

The submission is readily accessible, well-documented, and supports reproducibility

**Ethical Considerations:**

No, there are no or only very minor ethics concerns

**Final Justification:**

The authors have addressed my concerns through their rebuttal; accordingly, I reaffirm my ‘Accept’ rating and recommend to accept the paper .

**Limitations Weaknesses:**

1. The current benchmarks don't yet use all the data's features, like temporal information, or explore different learning methods like few-shot learning.
2. The method of finding cryptic groups based on user data means that lookalikes for very rare species might be missed.

**Strengths Contributions:**

CRYPTICBIO is the largest dataset of its kind by several orders of magnitude. It addresses the specific, under-resourced, and important challenge of cryptic species identification, moving beyond general-purpose biodiversity datasets. The paper contributes not just a dataset but also four well-designed benchmarks targeting common, unseen, endangered, and invasive species. The experiments provide a clear, impactful result: adding location data as a modality significantly boosts zero-shot classification performance on cryptic species.

---

> ### Author Rebuttal · Authors · 2025-07-25
>
> We sincerely thank the reviewer for the thoughtful and constructive feedback, as well as for recognizing the novelty and potential impact of our work. We are glad the reviewer found the CRYPTICBIO dataset, its multimodal nature, and the accompanying benchmarks and experimental insights compelling. In particular, we appreciate the emphasis on our contribution to the underexplored challenge of cryptic species identification, and the recognition of the value added by spatiotemporal context and multilingual vernacular data, a concern we detailed in the Appendix A.4 (lines 372-389).
>
> Below, we address the limitations raised:
>
> **"The current benchmarks don't yet use all the data's features, like temporal information, or explore different learning methods like few-shot learning."**
>
> We appreciate this observation and fully agree. At the same time, we emphasize that one of the key contributions of our work is showing that zero-shot models, when augmented with geographic and taxonomic context, already achieve significantly better performance on challenging cryptic species. Thus, even without additional labeled examples, multimodal context can compensate for fine-grained visual ambiguity. This highlights a major advantage in computational efficiency, which is increasingly important for sustainable and scalable biodiversity monitoring. Nonetheless, we acknowledge the value of additional temporal modality and few-shot evaluation and have noted this in the Limitations section (lines 228–232).
>
> **"The method of finding cryptic groups based on user data means that lookalikes for very rare species might be missed."**
>
> We appreciate this important observation. It is true that our current approach relies on observed misidentifications in iNaturalist's (i.e. "Similar Species" metadata), which are more likely to surface confusion patterns for frequently encountered species. As a result, visually similar but extremely rare species, which may have not yet have been recorded in iNaturalist observations to trigger systematic confusion, could be underrepresented in the cryptic groups. More broadly, rare species tend to be underrepresented in most community-sourced biodiversity datasets, including iNaturalist, due to lower encounter rates and limited geographic coverage. As a result, both the visibility and confusion of rare taxa are less likely to be captured through user activity alone. That said, our design prioritizes real-world, empirically observed confusion, as this reflects the practical challenges faced by users and models alike in field applications. Still, we fully agree that extending the coverage to include rare or under-reported cryptic lookalikes is valuable. In future work, we aim to complement community-sourced biodiversity data (i.e. citizen science data) with alternative sources such as taxonomic literature, expert-curated references, or model-based similarity graphs to capture rare but biologically plausible confusions. We have noted this in the Limitations section (lines 224–227).

---

> > ### Comment · Reviewer_szBE · 2025-08-09
> >
> > Thank you for your rebuttal, it solved most of my concerns and I will recommend accept this paper

---

> ### Comment · Area_Chair_UNs3 · 2025-08-08
> **Action Required: Respond to Rebuttals As Soon As Possible**
>
> Dear Reviewer szBE,
>
> It is less than 24h until the end of the Reviewer-Author Discussion period, and you still haven’t responded to the Author rebuttals. Please do so as soon as possible.
>
> It is bad practice to not participate in the discussion, especially when authors request it, and will be sanctioned as part of this year’s [Responsible Reviewing Initiative](https://blog.neurips.cc/2025/05/02/responsible-reviewing-initiative-for-neurips-2025/).
>
> Thank you,
>
> AC

---

### Author Response · Authors · 2025-08-08
**Planned revisions following rebuttal discussions**

We thank all reviewers for their valuable and constructive feedback. Below is a summary of all points that will be addressed in the revised version of the paper:

- Clarify distinction between 67K unique species and 52K cryptic groups (non-exclusive cryptic groups) in section 2;
- Update paper title to "CrypticBio: A Large Multimodal Dataset for Visually Confusing Species" to more accurately reflect the dataset's focus;
- Add to comparison Table 11:
  - AMI dataset
  - iNaturalist-2024 (iNat24)
  - Also, species overlap, % overlap, and Jaccard scores
- Add Top-1, Top-3, Top-5 accuracy (with 95% CI) + Precision, Recall, F1 scores in appendix E and also clarify macro vs weighted metrics (balanced dataset);
- Add discussions on at the Limitations in the main paper:
  - Impact of image quality (distance, blur, occlusion)
  - Why environmental variables reduce accuracy in zero-shot
  - Correlation between group size and performance (Spearman ~0.05)
  - Use of VLMs/LVMs for cryptic species detection (future work)
- Move cryptic group pipeline explanation from section 2 to section 3;
- Extend section 3 with full web scraping methodology;
- Update Git README with:
  - Quick-start guide
  - Environment setup instructions
  - Benchmark reproduction steps
  - Web scraper for cryptic group mining
- Add proper GBIF DOIs in References;
- Fix spelling and grammar of lines 154-157 “ENGENDERED”;

---

### Decision · Program_Chairs · 2025-09-18

**Decision:**

Accept (poster)

**Comment:**

**Paper Summary**

This paper presents CrypticBio, a large-scale multimodal dataset of 166 million images across 67,000 visually confusing ("cryptic") species. Curated from real-world misidentifications on iNaturalist, the dataset is enriched with spatiotemporal, taxonomic, and multilingual metadata. Benchmarks on the dataset demonstrate that including geographical context significantly improves zero-shot classification performance for these challenging species.


**Review Summary**

All four reviewers unanimously and confidently recommended acceptance. They praised the dataset's novelty, scale, and relevance for advancing fine-grained recognition in biodiversity. The method for mining cryptic groups from user data was highlighted as a key strength. Minor initial concerns regarding metrics, code documentation, reproducibility, and data citation were fully and satisfactorily addressed by the authors during the rebuttal, solidifying the consensus.


**Rationale for Recommendation**

I strongly recommend this paper for acceptance with a spotlight. It is a well-executed and valuable contribution that stands out for several reasons:
* Addresses a Critical Research Bottleneck: It tackles the critical, under-resourced challenge of classifying visually similar species, a key hurdle for real-world biodiversity AI.
* Excellent Scale and Multimodal Depth: The dataset is massive (166M images) and uniquely multimodal, incorporating location, time, and multilingual names, which enables new research directions.
* Clear Empirical Finding: The benchmarks provide a valuable and impactful insight: leveraging geographical context significantly boosts performance on this difficult task.
* Useful Resources: The dataset is accessible, and the authors have committed to releasing well-documented code to ensure full reproducibility, directly addressing reviewer feedback.

The authors' thorough rebuttal further confirmed the quality of this submission, which promises to become a valuable benchmark resource and to catalyze significant research in fine-grained and multimodal AI.